# M3Kang: Evaluating Multilingual Multimodal Mathematical Reasoning in Vision-Language Models

## Abstract

Despite state-of-the-art vision-language models (VLMs) have demonstrated strong reasoning capabilities, their performance in multilingual mathematical reasoning remains underexplored, particularly when compared to human performance. To bridge this gap, we introduce M3Kang, the first massively multilingual, multimodal mathematical reasoning dataset for VLMs. It is derived from the Kangaroo Math Competition, the world's largest mathematics contest, which annually engages over six million participants under the age of 18 across more than 90 countries. M3Kang includes 1,747 unique multiple-choice problems organized by grade-level difficulty, with translations into 108 culturally diverse languages, some of them including diagrams essential for solving them. Using this dataset, we conduct extensive benchmarking on both closed- and open-source SOTA models. We observe that, despite recent advances, models still struggle with basic math and diagram-based reasoning, with performance scaling with language presence and model size, but not with grade level. We also find that multilingual techniques can be effectively extended to the multimodal setting, resulting in significant improvements over baseline approaches. Our analysis also incorporates performance data from over 68,000 students, enabling direct comparison with human performance. We are open-sourcing M3Kang, including the English-only subset M2Kang, along with the framework and codebase used to construct the dataset.

## 1 Introduction

The rapid advancement of LLMs and VLMs has revolutionized artificial intelligence, unlocking unprecedented capabilities in reasoning as well as multilingual and multimodal processing. Yet, these models are typically evaluated on high-resource languages and text-only benchmarks, and in the instances where multilingual or multimodal evaluations have been conducted, findings reveal that they reason poorly in less commonly used languages (Shi et al., 2022a) and struggle to interpret diagrams accurately (Zhang et al., 2024). The lack of benchmarks at the intersection of these two dimensions not only limits our understanding of model robustness, but also poses a significant barrier to the development of inclusive and globally applicable AI systems.

To address this gap, we introduce M3Kang, a novel, highly multilingual, multimodal, mathematical benchmark, built from the Kangaroo Math Competition. Starting from a monolingual dataset, we propose a new pipeline for translating multimodal problems into multiple languages. Using this pipeline, we extend M3Kang to 108 languages selected to ensure linguistic diversity and varied online presence, therefore ensuring a representative and challenging evaluation landscape. By grounding the evaluation in mathematical reasoning—a domain that demands both linguistic precision and visual comprehension—we provide a high-fidelity testbed for assessing model generalization, cross-lingual consistency, and multimodal integration.

Using M3Kang, we conduct an exhaustive evaluation of top-performing VLMs, both open- and closed-source, with particular focus on cross-linguistic comparisons and the distinction between problems that require visual understanding and those that do not. To contextualize model performance, we also compare results against human baselines. Additionally, we assess state-of-the-art techniques for multilingual reasoning, to explore their applicability in the multimodal setting. The

code required to reproduce the experiments and generate the dataset is made available at `github.com/-----/M3Kang` and the dataset is available at `huggingface.com/-----/M3Kang`.

The main contributions of this paper are:

- **M3Kang**: A multilingual, multimodal benchmark dataset derived from the Kangaroo Math Competition, comprising 1,747 unique problems across multiple difficulty levels and supporting 108 languages, for a total of 111,198 problems after filtering.
- **A translation pipeline** for automatically creating multilingual and multimodal datasets from a monolingual source, including a framework to assess the quality of generated evaluations, along with its implementation.
- **An exhaustive evaluation** of leading VLMs across languages and inference techniques, and against human baselines, leading to a deeper understanding of SOTA model capabilities and limitations.

## 2 PREVIOUS WORK

**Multimodal, multilingual and mathematical benchmarks.** The rise of multimodal models with strong reasoning (OpenAI, 2024; Google, 2025; Anthropic, 2025) has driven the need for robust evaluation standards. In recent years, several multimodal benchmarks have emerged, including multimodal multidisciplinary ones like MMMU (Yue et al., 2024), ScienceQA (Lu et al., 2022) and MMT-Bench (Ying et al., 2024), as well as others focused on mathematical reasoning, such as MathVista (Lu et al., 2024), Math-Vision (Wang et al., 2024), MathVerse (Zhang et al., 2024) and OlympiadBench (He et al., 2024). Also, though trained for multilingual use, LLMs still perform best in English and struggle with low-resource languages (Raval et al., 2025). Consequently, multilingual benchmarks like Global-MMLU (Singh et al., 2025), XCOPA (Ponti et al., 2020), XTREME (Hu et al., 2020), and math-focused ones like MGSM (Shi et al., 2022a) and MMATH (Luo et al., 2025) have emerged to guide cross-lingual model development. However, few benchmarks assess the intersection of multilingual, multimodal and mathematical capabilities, and those that do focus on general knowledge rather than math and cover only a limited set of languages (see Table 1). The M3Kang benchmark, built on Kangaroo math problems, is designed to address this gap.

Table 1: Some similar datasets that contain multilingual, multimodal and mathematical problems. No translation technique indicates that the dataset consists of problems originally presented in their respective languages.

| Dataset | # of languages | # of problems | Translation technique | Unified pool | Topic |
|---|---|---|---|---|---|
| M3Kang | 108 | 111K | Automatic | Yes | Math |
| M3Exam (Zhang et al., 2023) | 9 | 12K | None | No | Varied |
| EXAMS-V (Das et al., 2024) | 11 | 20K | None | No | Varied |
| M5 (Schneider & Sitaram, 2024) | 41 | 237K | None | Partially | Varied |
| M4U (Wang et al., 2025) | 6 | 10005 | Automatic | Yes | Varied |
| PISA-Bench (Haller et al., 2025) | 6 | 732 | Automatic | Yes | Varied |
| PangeaBench (Yue et al., 2025) | 36 | 116K | Varied | Partially | Varied |

**Other Kangaroo Datasets.** Recently, Cherian et al. (2024) introduced SMART-840, a dataset consisting of 840 problems from the American Kangaroo competition. They also compared model and student performance, finding no positive correlation between human and AI capabilities in their dataset. Similarly, Sáez et al. (2025) introduced another dataset with Kangaroo problems from 4 different countries, and corroborated previous findings (Zhang et al., 2024) showing VLMs consistently underuse diagrams. However, different countries select different problems, so comparisons between languages with this dataset are unfair. In contrast to the datasets above, M3Kang is multilingual as it contains a common pool of questions in 108 languages, it contains both high and low-resource languages, and it is built using an open-source automated pipeline that facilitates easy expansion of the dataset to include additional problems and languages.

**Multilingual techniques.** While model performance varies across languages, new methods are proving effective in reducing these gaps. Shi et al. (2022b) observed that translating to English

with an external translator before applying CoT outperformed both CoT in the original language and English CoT without translation. Relatedly, Huang et al. (2024) found that including both the original question and its English translation in the prompt outperformed using only English. Qin et al. (2023) had similar results and proposed CLSP—applying this approach across multiple languages, aggregating with majority voting—which outperformed all other methods.

In parallel to these techniques, steering vectors (Subramani et al., 2022; Turner et al., 2024) have also been used to enhance multilingual capabilities of language models (Mahmoud et al., 2025; Zhang et al., 2025). While Mahmoud et al. (2025) steers representations toward English in a single layer, Zhang et al. (2025) shifts them to English and then back during generation with additional model training, both demonstrating the method's effectiveness.

**Challenges in visual reasoning**   While expert-level benchmarks grow (Phan et al., 2025; Glazer et al., 2025), LLMs still fail basic logic tasks (Wu et al., 2024), especially with visual problems (Huang et al., 2025). Even top models struggle with simple diagram-based math, sometimes performing better without visuals (Zhang et al., 2024)—a gap we also explore using M3Kang.

## 3   THE M3KANG BENCHMARK

Our dataset is built from problems in the Kangaroo Math Competition, the largest annual international contest for primary and secondary students. Each test includes 24 or 30 multiple-choice reasoning problems, organized by grade level and sometimes accompanied by figures (see Appendix A). We sourced data from the original PDFs of the 2007–2024 editions organized by the *Societat Catalana de Matemàtiques*, and processed it through a pipeline described below. The final dataset includes each problem in two formats: a text version of the question and an image containing the full problem (question, answers, and optional figure) with translations into the 108 languages described in Table 7. Text extraction enables text-based techniques and ensures evaluation focuses on reasoning rather than OCR.

In this section we provide the full explanation of the process from the raw monolingual PDF files into the full dataset with 108 languages (see Table 4 for a full list). Our framework includes three stages: (1) creating a clean annotated dataset from the source language (Catalan), (2) translating it to English, and (3) extending it to other languages. We open-source the full pipeline—including tools for parsing, annotation, translation, quality control, and deduplication—to support the creation of similar datasets in diverse linguistic and educational contexts.

### 3.1   OBTAINING THE BASE DATASET IN CATALAN

Figure 1 illustrates the construction of the initial Catalan dataset. We began by parsing the Kangaroo math problems in PDF format, converting them to images, and segmenting them into individual problems with extracted text. Bounding boxes were manually drawn around each question, excluding problems that featured Catalan-specific terms in images or answers to maintain linguistic neutrality. After cleaning parsing artifacts, the text was re-embedded into the image within the bounding box to preserve visual structure. A quality assurance pass using an LLM-as-a-judge flagged corrupted samples (samples with an incorrect bounding box or having their figure partially covered, for example), which were manually reviewed and corrected. Finally, duplicates were identified via textual similarity, and the version linked to the lowest educational level was retained.

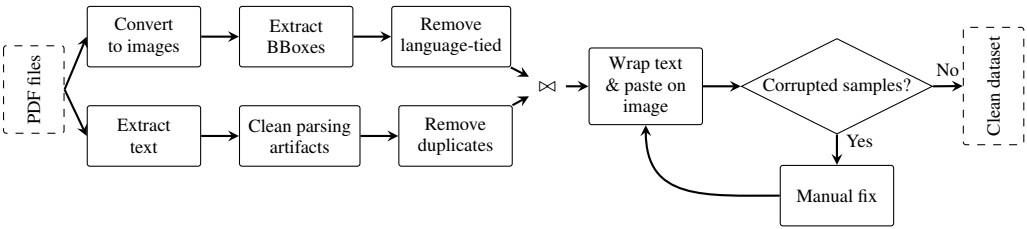

Figure 1: Pipeline to create the base dataset in Catalan

### 3.2 M2KANG: A MULTIMODAL MATHEMATICAL BENCHMARK IN ENGLISH

The second stage of the framework, shown in Figure 2, extends the Catalan dataset into English. This involves translating the question text from a source language $l_1$ = Catalan to a target language $l_2$ = English, and dynamically re-wrapping the translated text within the original bounding box to maintain the multimodal alignment. For translation to English, the availability of an LLM-as-judge enabled us to implement a corruption detection step (this time including translation issues), which was repeated to ensure the integrity of the newly generated samples. Any issues were manually corrected, and the process was iterated until the flagged examples were no longer incorrect.

Finally, we conducted a full manual review of the English dataset, with unclear cases double-checked by two reviewers, removing 42 mistranslations to ensure linguistically accuracy of the dataset and eliminate any bias from the Catalan origins (see Appendix B.2). We also open-source the English benchmark subset independently (M2Kang), which can be used to evaluate non-multilingual, multimodal reasoning in VLMs.

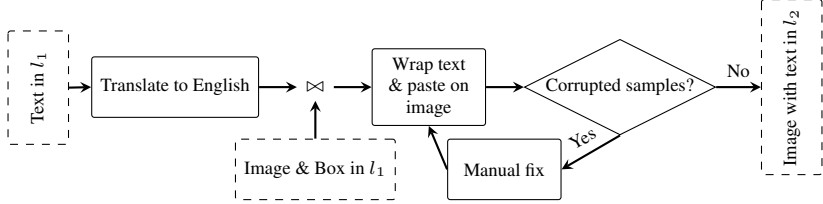

Figure 2: Pipeline to create the dataset in English

### 3.3 CREATING A MULTIMODAL MULTILINGUAL DATASET

The process to obtain a multilingual version is described in Figure 3. The main differences with the pipeline for the English dataset is that we do not need to check for corrupted samples anymore (as it has been done extensively), but we have to add a step to determine which samples are mistranslated to discard them. To assess the quality of translated problems without relying on ground truth references, we adopt a backtranslation-based evaluation strategy, described in Section 3.3.1, and discard mistranslated samples. The result is a robust, multimodal dataset that supports multilingual applications, with each image-text pair carefully curated for clarity, consistency, and relevance.

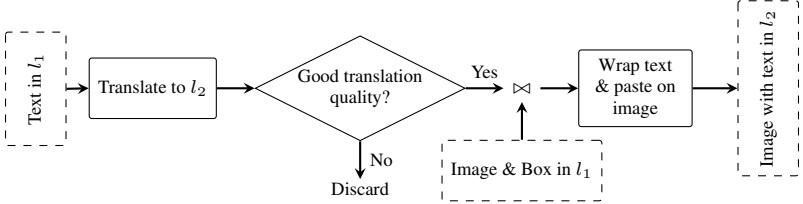

Figure 3: Pipeline to translate dataset from language $l_1$ (source) to $l_2$ (target)

### 3.3.1 CHECKING TRANSLATION QUALITY

To ensure the quality of the translated samples, we employ reference-free machine translation quality estimation methods, with the requirement that these methods be available for a large amount of language pairs. The latter requirement reduces the search space, as SOTA methods like YiSi-2 (Lo & Larkin, 2020) or COMET (Rei et al., 2020) are model-dependent and, hence, inherit the limitations of said models regarding underrepresented languages such as Maltese or Tsonga. For this reason, we consider using backtranslation-based metrics, as these depend only on the presence of a backtranslator for the language of choice, which is a possibility for a large amount of languages via NLLB (NLLB Team et al., 2024). Figure 4 shows the workflow we use to implement the backtranslation-based metric approach.

After validation experiments using the Flores-200 dataset (NLLB Team et al., 2022), we opted for the max chrf score between the source and the backtranslations, as it correlated well with the chrf++ score (Popović, 2017) and is robust to mistranslation by a subset of backtranslation models, and used a threshold of $T = 0.625$. More details on the metric selection and validation can be found in Appendix B.3 and Appendix B.4. Note that this translation quality check implies that some problems are not translated to certain languages, see Table 4 in Appendix B.1 for more information on the number of remaining problems after the quality check and Appendix C.3 for the implications on multilingual evaluation.

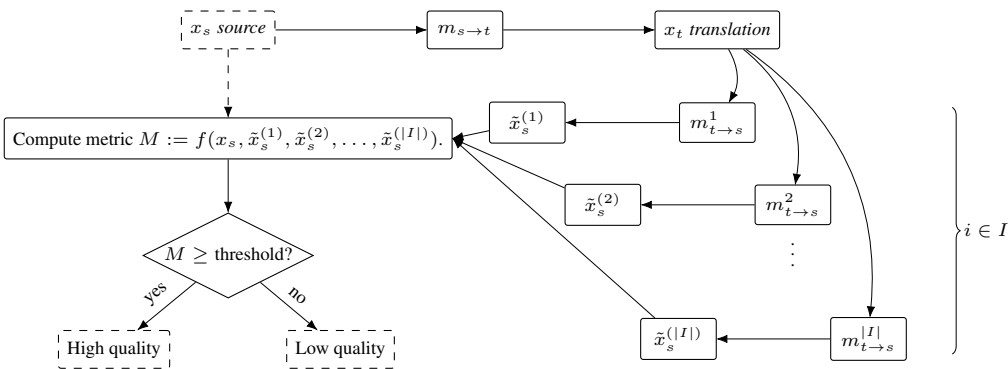

Figure 4: Pipeline to compute backtranslation-based translation quality metrics

# 4 EXPERIMENTS

Our benchmark offers a unique opportunity to study reasoning at the intersection of **multimodality** and **multilinguality**. While these dimensions have traditionally been explored in isolation, we aim to understand how patterns observed in one domain might generalize to the other. Throughout this section, we benchmark known open and closed models on M3Kang, study the relationship between human-labeled difficulty and model performance, as well as evaluate whether techniques that have shown to improve multilingual performance transfer to the multimodal scenario. We also study the performance gap across text-only problems and problems including figures. To do so, we select 16 languages from the original set of 108, ensuring a balanced mix of low-, mid-, and high-resource languages based on their internet presence, so as to cover different points along the resource spectrum (see Appendix C.2 for more information). The experiments we run throughout this section have been performed with the following models: Gemini-2.5-Pro, Claude-4-Sonnet, GPT-4.1, GPT-4o, Kimi-Thinking-2506, Gemma-12B, Qwen-2.5-7B, Phi-4, Gemma-4B, QWen-2.5-3B, MoonDream and Gemma-1B (see Appendix C.1 for more details on the choice of models).

In our experiments, to prompt the VLMs to answer a question, we use a system prompt that leverages Chain-of-Thought (Wei et al., 2023) and instructs the model to answer in a simple format, demanding reasoning traces after `Reasoning:` and the answer following `Answer:` in the format `A),B),C),D)` or `E)`. The prompts were translated in the language of the dataset for benchmarking in languages other than English (see Appendix D for the exact prompts). To parse the response, we select the latest appearance of any answer string, and an `N` if none is found, indicating no answer.

## 4.1 MODEL PERFORMANCE ON M3KANG

We evaluate model performance on M3Kang using the average accuracy over three independent runs. To ensure fairness across languages, we report results on the subset of correctly translated samples for each language, which is justified as the variation in accuracy across different correctly translated subsets is not statistically significant for the vast majority of models (see Appendix C.3 for details). For the Gemma family, we only executed the benchmark in English, as we saw the performance was worse than random guessing. This is due to the model not adhering to the required output format specified in the prompt, a limitation we also encountered with MoonDream.

Table 2: Model accuracy across models and Internet language resource clusters (in %).

| Model name | English | Average | High-res. | Mid-res. | Low-res. |
|---|---|---|---|---|---|
| Gemma-1B | $15.3 \pm 0.8$ | - | - | - | - |
| MoonDream | $8.9 \pm 0.7$ | $12.1 \pm 0.2$ | $11.9 \pm 0.3$ | $12.1 \pm 0.6$ | $12.7 \pm 0.5$ |
| Qwen-2.5-3B | $30.3 \pm 1.0$ | $24.1 \pm 0.3$ | $25.7 \pm 0.4$ | $25.6 \pm 0.7$ | $19.4 \pm 0.5$ |
| Gemma-4B | $18.8 \pm 0.7$ | - | - | - | - |
| Phi-4 | $28.8 \pm 1.1$ | $23.0 \pm 0.3$ | $23.7 \pm 0.4$ | $22.9 \pm 0.7$ | $21.3 \pm 0.6$ |
| Qwen-2.5-7B | $40.0 \pm 1.0$ | $29.3 \pm 0.3$ | $32.1 \pm 0.4$ | $30.1 \pm 0.7$ | $22.0 \pm 0.5$ |
| Gemma-12B | $18.8 \pm 0.7$ | - | - | - | - |
| Kimi-Thinking-2506 | $50.5 \pm 1.0$ | $40.7 \pm 0.3$ | $45.8 \pm 0.4$ | $46.7 \pm 0.7$ | $24.8 \pm 0.4$ |
| GPT-4o | $43.7 \pm 1.0$ | $40.7 \pm 0.3$ | $41.5 \pm 0.4$ | $43.7 \pm 0.7$ | $36.7 \pm 0.6$ |
| Claude-4-Sonnet | $67.1 \pm 1.0$ | $62.3 \pm 0.3$ | $63.3 \pm 0.4$ | $63.7 \pm 0.8$ | $59.2 \pm 0.6$ |
| Gemini-2.5-Pro | $76.4 \pm 0.9$ | $75.6 \pm 0.3$ | $76.2 \pm 0.4$ | $73.5 \pm 0.7$ | $75.4 \pm 0.5$ |

Among open models, we see that performance averaged in high-resource languages is lower than performance in English, and performance in low-resource languages is often close to random. Kimi is the strongest among the open-source models, and bests GPT-4o in English performance. Among closed models, we observe that performance is better than random across all language resources. Gemini-2.5-Pro bests all categories for the closed-source models evaluated, with Claude-4-Sonnet behind and GPT-4o at the bottom. Notably, even though GPT-4o is less performant in English than Kimi-Thinking-2506, it is still better on average and across low resource languages, indicating that it is a stronger multimodal reasoner in low-resource language contexts.

## 4.2 LANGUAGE COMPARISON

In this section, we provide a more fine-grained analysis of multilingual performance. This analysis leverages that M3Kang includes a wide range of languages with diverse origins and varying levels of Internet presence (see Appendix C.2 for details). Out of the 16 languages available, we determined that 12 could be fairly compared (see Appendix C.3 for details). The average accuracy across all models is shown in Figure 5, alongside Spearman's correlation coefficients between accuracy and Internet presence (see Table 7 for exact values).

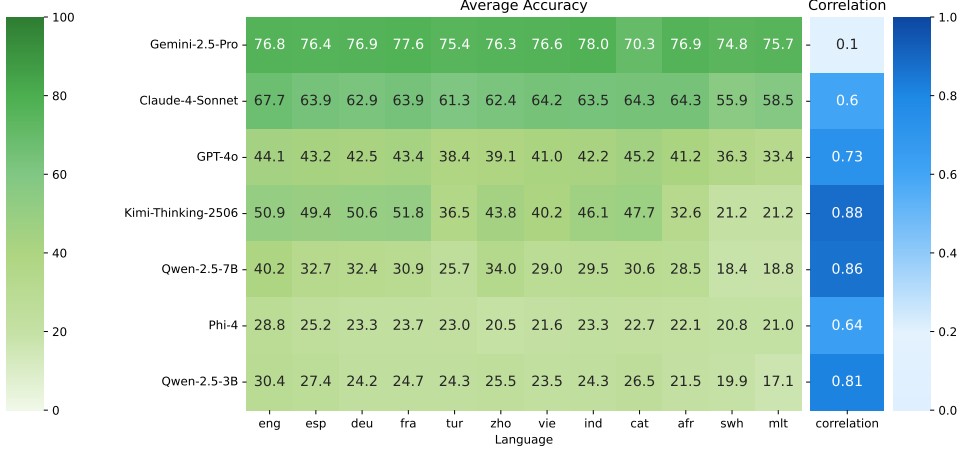

Figure 5: In green, average accuracy of VLMs across culturally diverse languages sorted by their Internet presence. In blue, for each VLM, Spearman's correlation coefficient between model accuracy and Internet presence.

Our analysis reveals that, for all models except Gemini-2.5-Pro, the average accuracy per language strongly correlates with its Internet presence, highlighting the impact of training data imbalances on multilingual performance. Notably, Gemini-2.5-Pro demonstrates robust multilingual capabilities, maintaining English-level accuracy across a wide range of languages. In contrast, Kimi-Thinking-

2506 performs well in English and closely related languages but shows a significant drop in performance for less common or typologically distant languages. Spearman's coefficients further indicate that the correlation between performance and Internet presence is stronger in smaller models, likely due to their limited capacity to generalize across multiple languages simultaneously.

## 4.3 MULTILINGUAL TECHNIQUES

In this section, we address two key questions that arise when moving from text-only to multimodal reasoning benchmarks such as M3Kang: **(1)** Do techniques that enhance multilingual performance in text-based settings remain effective in a multimodal context? **(2)** Among these techniques, which perform best under a constrained memory budget?

To investigate these questions, we use the Qwen-2.5 family of models as reasoners and compare three multilingual reasoning strategies against a baseline with no additional technique (vanilla). The three strategies are: **Model Tongue Reasoning (MTR)**, consisting in translating the question to English and prompting the reasoner with both versions (Huang et al., 2024); **CLSP**, which applies MTR across multiple languages and aggregates the outputs via majority voting (we use the four most commonly used Latin-based languages on the Internet: English, Spanish, German, and French); and **Activation Steering**, from Mahmoud et al. (2025) and Zhang et al. (2025), which modifies internal activations $h^l$ to improve multilingual alignment ($\hat{h}^l = h^l + c \cdot z_{o \to en}^l$ in early layers and $\hat{h}^l = h^l + c \cdot z_{en \to o}^l$ in later ones), using means of activations to compute the $z$ vectors, as in Turner et al. (2024). Further experiments optimizing the Activation Steering technique are provided in Appendix C.4.

All three techniques leverage the fact that VLMs exhibit their strongest capabilities in English, due to the language distribution in their training data. For methods requiring translation (MTR and CLSP), we employ the NLLB family of models (NLLB Team et al., 2022) at three scales: 600M, 1.3B, and 3.3B parameters. To ensure fair comparisons, we include the translator's parameters in the total model size. Figure 6 summarizes the results of these experiments.

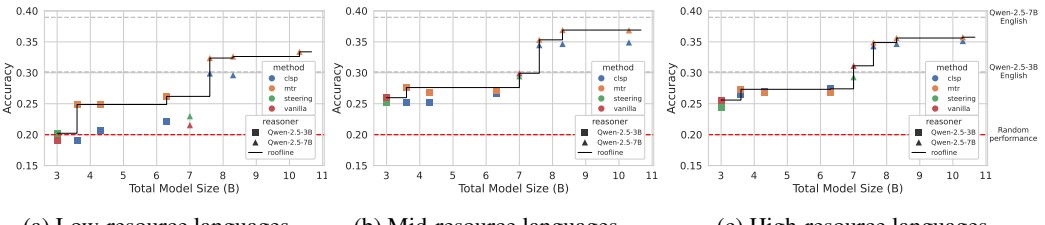

(a) Low-resource languages      (b) Mid-resource languages      (c) High-resource languages

Figure 6: Performance of multilingual techniques versus a vanilla baseline using the Qwen-2.5 family (reasoners) and NLLB models (translators), plotted against total parameter count. For MTR and CLSP, each reasoner has three points corresponding to NLLB sizes (600M, 1.3B, 3.3B). Horizontal lines indicate the random baseline and performance on English data. The rooftop line marks the best-performing technique for each parameter budget and scores are averaged across language resource level.

We find that MTR consistently achieves the strongest overall performance. In contrast, CLSP underperforms despite being designed as an enhancement of MTR. Activation Steering uses fewer parameters than both MTR and CLSP, but it only outperforms the vanilla baseline in low-resource languages and its gains remain modest compared to MTR: on Qwen-7B, its performance remains far below the roofline established by MTR on Qwen-3B. Remarkably, MTR substantially reduces performance disparities across resource levels. For instance, the gap between low- and high-resource settings in the vanilla model is 7.1% and 12.3%, whereas with MTR it narrows to 2.5% and 3.1% for Qwen-2.5-3B and Qwen-2.5-7B, respectively. Moreover, MTR achieves performance levels approaching those observed for English. This shows that the most substantial improvements overall occur in smaller models and low-resource languages, where language understanding is the primary limitation rather than reasoning ability. Finally, it is worth noting that using a stronger translator model generally improves performance, though the gains are relatively modest. This suggests that,

in low-budget scenarios, smaller translators can be employed without incurring a significant drop in capabilities.

## 4.4 Text-only vs Figure problems

To assess modality sensitivity, we compared model performance on text-only versus figure-based problems in M2Kang (English). Consistent with prior work (Zhang et al. (2024), Huang et al. (2025)), Figure 7 reveals a clear and consistent gap: all capable models (i.e., those above the English random baseline) achieve substantially higher accuracy on text-only problems than on figure-based ones, both overall and at every level (right panel). We also examined human performance (left panel). Interestingly, humans find figure-based problems easier than text-only, whereas VLMs show the opposite trend. This discrepancy persists across model families and sizes, suggesting that current VLMs still struggle to integrate visual information effectively for mathematical reasoning. This reinforces our claim that multimodal questions are significantly harder for VLMs than text-only counterparts of similar difficulty, and that the difference is not attributable to easier problems. Despite recent improvements in text-based reasoning, the figure modality introduces additional complexity (such as spatial relationships and diagram interpretation) that current architectures and training regimes do not fully capture. Overall, these results underscore the need for better visual interpretation to close the gap between text and figure understanding in mathematical contexts.

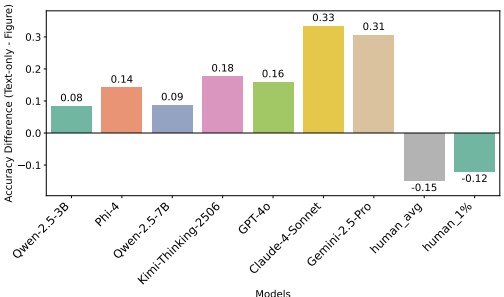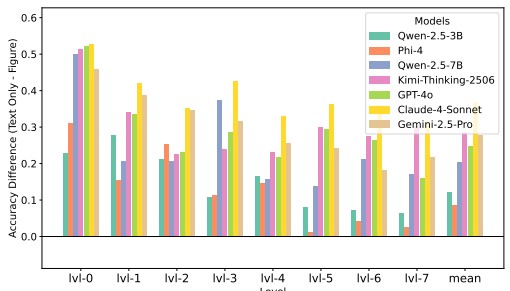

Figure 7: Difference in accuracies between two problem types: text-only minus figure-based. (Left) Differences for the 2024 subset, including human performance. human_avg represents the average of all participants, and human_1% the average of the top 1%. While participants found figure-based problems easier, models still perform better on text-only problems. (Right) Differences across all levels and overall average in the entire dataset. All models consistently perform better on text-only problems compared to figure-based problems at every level.

We complement this analysis of problem modality with an analysis based on problem category in Appendix C.6.

## 4.5 Humans vs VLMs

The *Societat Catalana de Matemàtiques* provided problem-level performance data for over 68,000 participants aged 10 to 18 who took part in the 2024 Kangaroo Mathematics Competition in Catalonia. The dataset spans 114 problems and enables a detailed comparison between human and model performance. In this section, we examine the degree to which human performance aligns with that of computational models and explore whether the difficulty perceived by human participants corresponds to the difficulty estimated by these models. We analyze the performance of Kimi-Thinking-2506 in detail and refer to Appendix C.5 for analysis of other models.

Table 3 reveals two notable behaviors of Kimi-Thinking-2506. First, as previously discussed in Section 4.1, model performance declines markedly with decreasing Internet presence of the language. Quantitatively, the drop from high- to low-resource languages corresponds to a reduction of approximately 40–60 percentile points. In practical terms, this shift can represent a transition from top-of-the-class performance to that of an average or below-average student. Second, the model's performance does not decrease monotonically with increasing problem difficulty (according to human understanding of difficulty). Kimi-Thinking-2506 ranks above the 94th percentile on problems classified as levels 3-6, yet only above the 44th percentile on the lower levels. These behaviors are

Table 3: Percentile ranks that Kimi-Thinking-2506 would have achieved on the 2024 test, segmented by problem difficulty level and language classes based on Internet presence. See Appendix C.5.1 for additional percentile ranks on other models.

| Kimi's Percentile (↑) | Lvl 0 | Lvl 1 | Lvl 2 | Lvl 3 | Lvl 4 | Lvl 5 | Lvl 6 | Lvl 7 |
|---|---|---|---|---|---|---|---|---|
| English | 47.3% | 57.5% | 73.8% | 96.9% | 95.3% | 94.5% | 95.4% | 78.1% |
| High-resource langs. | 36.8% | 56.3% | 64.4% | 93.6% | 91.2% | 89.3% | 88.1% | 66.2% |
| Mid-resource langs. | 36.8% | 57.5% | 67.9% | 93.6% | 91.2% | 90.6% | 88.1% | 67.7% |
| Low-resource langs. | 11.0% | 14.5% | 19.2% | 41.5% | 33.7% | 24.0% | 18.0% | 21.2% |

present across models (see Appendix C.5). Based on the findings in Section 4.1, an attentive reader might infer that lower-level problems contain a higher proportion of image-based questions—and this is indeed the case (see Figure 15). However, this alone does not fully account for the observed performance gap. While lower levels do feature more image-based questions, Figure 8 uncovers a counterintuitive trend: most models perform better on image-based problems at higher difficulty levels, with Gemini-2.5-Pro exhibiting the strongest correlation. Manual inspection reveals that lower-level image-based problems focus mainly on visual understanding, while higher-level ones embed the image in the text and require more abstract reasoning. Unlike humans, modern models (often trained on olympiad-style geometry tasks) can increasingly perform such reasoning without relying on the image itself. Conversely, as expected, performance on text-only questions tends to decline with increasing difficulty (see Figure 16).

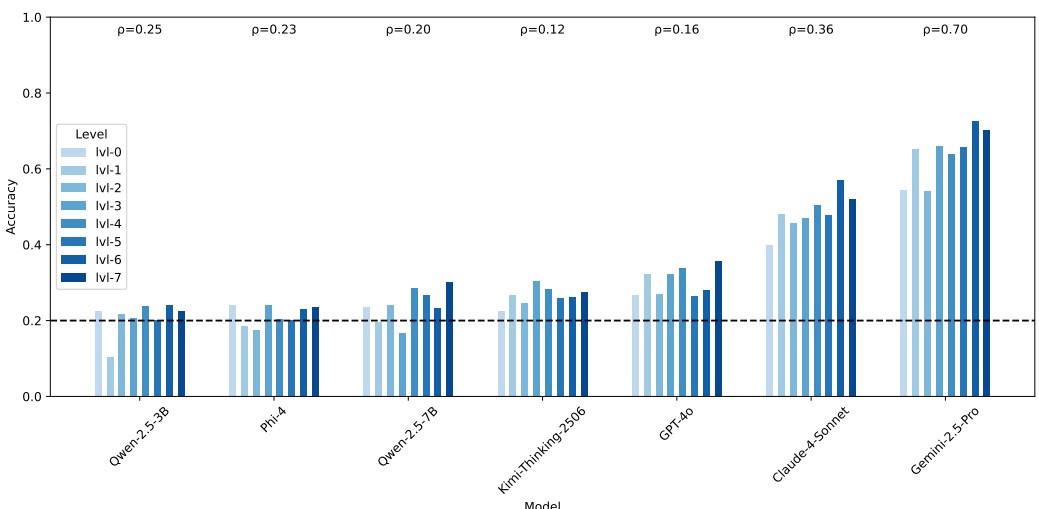

Figure 8: Accuracy of problems that contain figures across levels and models, with corresponding Spearman's correlation coefficients accuracy vs level.

Finally, following Cherian et al. (2024), we computed advanced metrics such as the Difficulty Index, Discriminative Index, and Weight Correlation. However, our analysis does not replicate the trends observed in their study (see Table 10 in Appendix C.5). Overall, our findings suggest no significant correlation between the reasoning capabilities of AI models and those of young students, indicating that their problem-solving approaches may be rooted in fundamentally different modes of reasoning. In terms of future research directions, this points to the fact that if we want to evaluate VLMs on more human-like problem-solving skills, future benchmarks should include multi-step reasoning tasks, combine visual interpretation with symbolic inference and assess intermediate reasoning steps rather than only final outputs. Such benchmarks would better capture the integrated reasoning strategies humans employ and provide a more meaningful measure of progress toward human-aligned capabilities.

## 5 LIMITATIONS AND FUTURE WORK

While our study lays a strong foundation, it also has certain limitations that suggest promising directions for future work:

- The automatic translation pipeline introduces uneven problem amounts across languages, which could be improved with stronger models or human translations, though the latter would greatly increase complexity.
- We evaluated some of the existing VLMs and multilingual reasoning techniques, excluding those that involve fine-tuning; broader comparisons would provide a more complete picture.

## 6 CONCLUSION

We introduced M3Kang, the first massively multilingual, multimodal reasoning dataset, covering 108 languages with a total of 111,198 problems. To achieve this scale, we designed an automated translation pipeline that leverages a back-translation-based quality metric, ensuring that only sufficiently accurate translations are retained while maintaining broad coverage for benchmarking. This pipeline not only facilitated the creation of M3Kang but can also serve as a foundation for building future multilingual datasets.

Using M3Kang, we conducted extensive benchmarking that yielded several key insights. We confirmed prior findings that, despite progress on advanced benchmarks, state-of-the-art models still struggle with basic mathematics and logic, particularly when diagrams are involved. Among tested models, Gemini-2.5-Pro consistently achieved the best performance across all languages, while open models showed a strong correlation between size and accuracy, except for the Gemma family and Moondream, which performed worse than random, likely due to limited reasoning training. We also observed that language accuracy correlates with Internet presence, especially for smaller models, whereas top closed models exhibit near language-agnostic reasoning, underscoring the benefits of scale for abstraction. Finally, we demonstrated that multilingual techniques can be effectively applied in multimodal settings, with MTR outperforming more complex methods despite its simplicity.

## 7 REPRODUCIBILITY STATEMENT

This work has been developed with a strong emphasis on openness and reproducibility, including the dataset creation pipeline, the full configuration details for every experiment, and the exact code used to generate each figure and plot. We will make all materials available via Hugging Face and Github after the rebuttal phase.

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

## A    EXAMPLE OF KANGAROO PROBLEM

26. Leo has drawn a closed path on a rectangular prism and has unfolded it. Which unfolding cannot be Leo's?

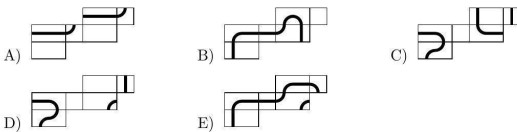

26. En Leo ha dibuixat un camí tancat en un prisma rectangular i l'ha desplegat. Quin desplegament no pot ser el d'en Leo?

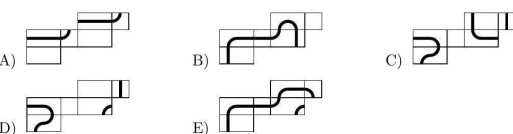

Leo ha dibujado un camino cerrado en un prisma rectangular y lo ha desplegado. ¿Qué desplegamiento no puede ser de Leo?

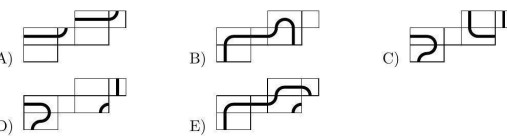

26. Leo hat einen geschlossenen Pfad auf ein rechteckiges Prisma gezeichnet und ihn entfaltet. Welche Entfaltung kann nicht von Leo sein?

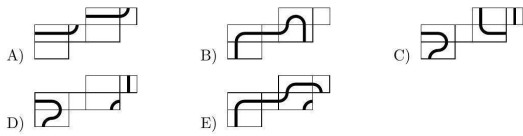

26. Leo đã vẽ một đường đi khép kín trên một lăng kính hình chữ nhật và đã mở ra nó.

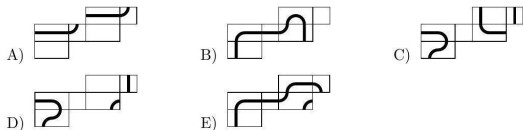

Leo amechora njia iliyofungwa kwenye prisma ya mstatili na ameifungua. Ni ufunguzi gani hauwezi kuwa wa Leo?

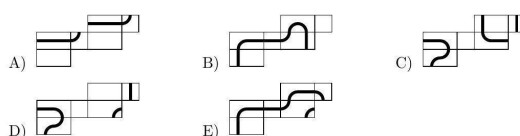

Figure 9: Example of an M3Kang problem that requires visual understanding. In English, Catalan, Spanish, German, Vietnamese and Swahili (from top to bottom). The problem corresponds to problem 26 of 2023 at level 6.

## B    FURTHER INFORMATION ABOUT THE DATASET GENERATION PROCESS

### B.1    LANGUAGES IN M3KANG

Below, we present the complete list of the 108 languages included in the M3Kang dataset, along with the number of samples for each language that met the quality criteria described in Section 3.3.

### B.2    MANUAL ENGLISH TRANSLATION VERIFICATION

To complement the automated translation pipeline, we performed a dedicated human translation check on the Catalan-to-English dataset. The objective of this process was twofold: (i) to ensure that the translated English samples were linguistically accurate and faithfully conveyed the original

Table 4: Number of samples per language in M3Kang. 1747 is the maximum amount of samples possible, which corresponds to the number of samples in English and Catalan, and the total number of samples is 111,198.

| Language | Samples | Language | Samples | Language | Samples |
|---|---|---|---|---|---|
| English | 1747 | Sundanese | 1373 | Silesian | 831 |
| Catalan | 1747 | Czech | 1372 | Pangasinan | 776 |
| French | 1740 | Lombard | 1344 | Tswana | 760 |
| Esperanto | 1736 | Occitan | 1325 | Tsonga | 727 |
| Portuguese | 1703 | Turkish | 1324 | Ligurian | 727 |
| Tagalog | 1691 | Polish | 1323 | Samoan | 721 |
| Spanish | 1689 | Hausa | 1296 | Plateau Malagasy | 655 |
| Arabic | 1686 | Limburgish | 1273 | Crimean Tatar | 646 |
| Standard Malay | 1660 | Croatian | 1270 | Basque | 589 |
| Maltese | 1653 | Friulian | 1268 | Kabuverdianu | 586 |
| Italian | 1650 | North Azerbaijani | 1263 | Latgalian | 584 |
| Venetian | 1649 | Icelandic | 1263 | Asturian | 568 |
| Afrikaans | 1647 | Bosnian | 1253 | Kinyarwanda | 551 |
| Indonesian | 1624 | Northern Kurdish | 1216 | Balinese | 490 |
| Swahili | 1615 | Zulu | 1216 | Lingala | 417 |
| Tosk Albanian | 1608 | Xhosa | 1174 | Buginese | 353 |
| Norwegian Bokmål | 1601 | Waray | 1168 | Rundi | 338 |
| Haitian Creole | 1591 | Slovenian | 1166 | Guarani | 317 |
| Cebuano | 1585 | Ilocano | 1147 | Turkmen | 300 |
| Sicilian | 1578 | Japanese | 1146 | Yoruba | 293 |
| Irish | 1572 | Southern Sotho | 1116 | Ganda | 265 |
| Romanian | 1536 | Lithuanian | 1068 | Swati | 247 |
| Galician | 1520 | Estonian | 1062 | Mizo | 233 |
| Swedish | 1514 | Sardinian | 1061 | Fijian | 219 |
| Papiamento | 1510 | Finnish | 1030 | Ewe | 187 |
| Norwegian Nynorsk | 1461 | Faroese | 1008 | Kabyle | 159 |
| Slovak | 1461 | Hungarian | 999 | Nigerian Fulfulde | 152 |
| German | 1455 | Northern Uzbek | 996 | Tamasheq | 131 |
| Dutch | 1448 | Somali | 988 | Central Kanuri | 124 |
| Vietnamese | 1442 | Nyanja | 979 | Twi | 117 |
| Scottish Gaelic | 1438 | Shona | 963 | Akan | 114 |
| Javanese | 1438 | Northern Sotho | 932 | Bambara | 113 |
| Chinese (Simplified) | 1427 | Maori | 908 | West Central Oromo | 113 |
| Korean | 1395 | Minangkabau | 882 | Tok Pisin | 113 |
| Luxembourgish | 1381 | Igbo | 865 | Nuer | 112 |
| Banjar | 1375 | Standard Latvian | 848 | Tumbuka | 112 |

meaning, and (ii) to verify that no systematic bias was introduced as a consequence of the Catalan source material.

Each translated sample was independently reviewed and assigned a quality label on a four-point scale, based solely on the comparison between the original Catalan text and its English translation:

- **Label 1: Unusable.** The translation was incorrect to the extent that the sample could not be meaningfully retained.
- **Label 2: Ambiguous.** The translation was unclear, incomplete, or overly dependent on the corresponding image to be interpretable.
- **Label 3: Imperfect but usable.** Minor translation errors were present, yet the intended problem or meaning was preserved.
- **Label 4: Perfect.** The translation was fully accurate, fluent, and faithful to the source text.

To reduce subjectivity and strengthen reliability, all samples initially labeled as 1 or 2 underwent a second round of evaluation by an independent human reviewer. This step ensured that potentially usable samples were not discarded prematurely and that unusable ones were consistently identified.

Through this multi-stage review, we identified a total of 42 mistranslations. These samples were removed from the dataset to uphold its overall quality. The remaining samples, particularly those labeled as 3 or 4, were retained, thereby ensuring that the dataset is both linguistically robust and representative of the original Catalan material without introducing translation-induced bias.

This human verification process highlights the importance of combining automated translation pipelines with targeted manual checks. While automated methods enable scalability across many languages, human review provides an additional safeguard in cases where translation quality directly impacts downstream tasks. By prioritizing accuracy, we ensure that the English dataset derived from Catalan maintains both reliability and fairness.

### B.3 REFERENCE-FREE MACHINE TRANSLATION QUALITY METRIC SELECTION

#### B.3.1 VALIDATION METHODOLOGY

In order to select amongst different backtranslation based metrics, we compute backtranslations on the Flores-200 (NLLB Team et al., 2022) dataset in Spanish, French, German, Swahili and Vietnamese, which we selected for their compatibility with other SOTA methods such as BERTScore (Zhang et al., 2020) or COMET (Rei et al., 2020). We then computed standard MT reference-based metrics such as BLEU (Papineni et al., 2002), chrf++ (Popović, 2017) and ROUGE (Lin, 2004) and checked the Spearman correlation with different candidate backtranslation metrics.

#### B.3.2 METRICS CONSIDERED

We tested semantic metrics, such as the proportion of backtranslations that are semantically equivalent to the source, with DeBERTa (He et al., 2021) used for entailment, as well as the mean, max and min of F-1, precision and recall of BERTScore across backtranslations. We also tested a series of syntactic metrics, including the mean, max and min of BLEU, chrf++ and ROUGE-1 (precision, recall and F-1) across backtranslations. Additionally, we added COMET for reference. For backtranslations, we used NLLB 3.3B (NLLB Team et al., 2022), Emma-500 (Ji et al., 2025), Madlad (Kudugunta et al., 2023) and WiNGPT-Babel-2 (winninghealth, 2025).

Some selected results can be found in Figure 10.

#### B.3.3 RESULTS

We observed that $\max_{i \in I} \mathrm{chrf++}(x_s, \tilde{x}_s^{(i)})$ correlates with $\mathrm{chrf++}(x_s, x_{ref})$ in all languages tested (Spearman's $\rho \in [0.15, 0.42]$). Table 5 shows the correlation to be statistically significant at 95% for all languages. We also observed good correlation of $\mathrm{mean}_{i \in I} \mathrm{chrf++}(x_s, \tilde{x}_s^{(i)})$, COMET and $\mathrm{mean}_{i \in I} \mathrm{BERTScore\text{-}precision}(x_s, \tilde{x}_s^{(i)})$ with reference-based metrics, but selected the max-based metric over the mean-based one for its robustness in very low resource languages.

Table 5: Spearman correlation and $p$-value for the test (probability of generating such dataset with 0 Spearman correlation) between $\max_{i \in I} \mathrm{chrf++}(x_s, \tilde{x}_s^{(i)})$ and $\mathrm{chrf++}(x_s, x_{ref})$.

| Target translation language | Spearman's $\rho$ | $p$-value |
|---|---|---|
| Spanish | 0.15 | $3.162 \times 10^{-6}$ |
| German | 0.42 | $3.754 \times 10^{-45}$ |
| Vietnamese | 0.29 | $1.039 \times 10^{-20}$ |
| Swahili | 0.38 | $1.102 \times 10^{-36}$ |
| Turkish | 0.36 | $7.848 \times 10^{-33}$ |
| French | 0.31 | $2.696 \times 10^{-24}$ |

In order to select the threshold, we look at L1 regression slope (to avoid outliers) and see that chrf++$(x_s, x_{ref})$ and $\max_{i \in I}$ chrf++$(x_s, \tilde{x}_s^{(i)})$ follow essentially the $y = 0.8x$ line, so it suffices to choose a good threshold for chrf++$(x_s, x_{ref})$ and divide it by 0.8. We arbitrarily select 0.5 chrf++ as a threshold for a good translation, as this is the state of the art in Machine translation for high resource languages (NLLB Team et al., 2022), and hence our threshold for $\max_{i \in I}$ chrf++$(x_s, \tilde{x}_s^{(i)})$ is $\frac{0.5}{0.8} = 0.625$. We therefore discard any sample with $\max_{i \in I}$ chrf++$(x_s, \tilde{x}_s^{(i)}) < 0.625$, and deem it mistranslated.

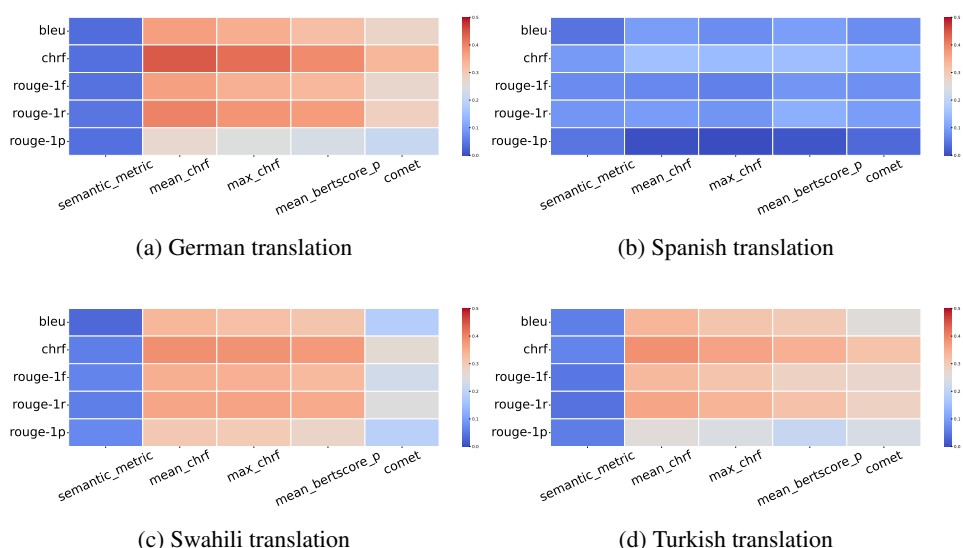

(a) German translation          (b) Spanish translation

(c) Swahili translation          (d) Turkish translation

Figure 10: Spearman correlation heatmap slices across selected metrics for four translation languages, $y$-axis having reference-based metrics and $x$-axis having reference-free metrics. Higher (more red) is better.

### B.4 HUMAN VALIDATION OF AUTOMATIC METRIC RELIABILITY

To ensure that our selected automatic metric is a reliable proxy for translation quality, we conducted a manual evaluation on a subset of the data. Specifically, we randomly sampled 30 problems in Spanish, 30 in French, 40 in Korean and 40 in Basque, ensuring a near-uniform distribution across the automatic metric range. Each sample was manually scored by human annotators on a scale from 0 to 10, where scores below 5 indicate that the translation should not be used.

Figure 11 shows the relationship between human scores and the automatic metric for the three languages. The observed Pearson correlations of 0.8722 (Spanish), 0.8475 (French), 0.7276 (Korean) and 0.7939 (Basque) are remarkably high, especially considering that human scores tended to be bimodal (favoring very low or very high ratings), while the automatic metric was more uniformly distributed. This inherent difference in score distributions typically reduces correlation, making these results even more compelling.

Importantly, we observed an extremely low number of false positives—cases where the automatic metric suggested a good translation but human evaluators disagreed. This is critical for our pipeline, as false positives represent mistranslations that could erroneously be included in the dataset. While some false positives are inevitable in a non-human-based pipeline, the rarity of such cases in our validation strongly supports the reliability of our metric.

These findings confirm that $\max_{i \in I}$ chrf++$(x_s, \tilde{x}_s^{(i)})$ is a robust and trustworthy metric for filtering translations in our multilingual dataset construction pipeline.

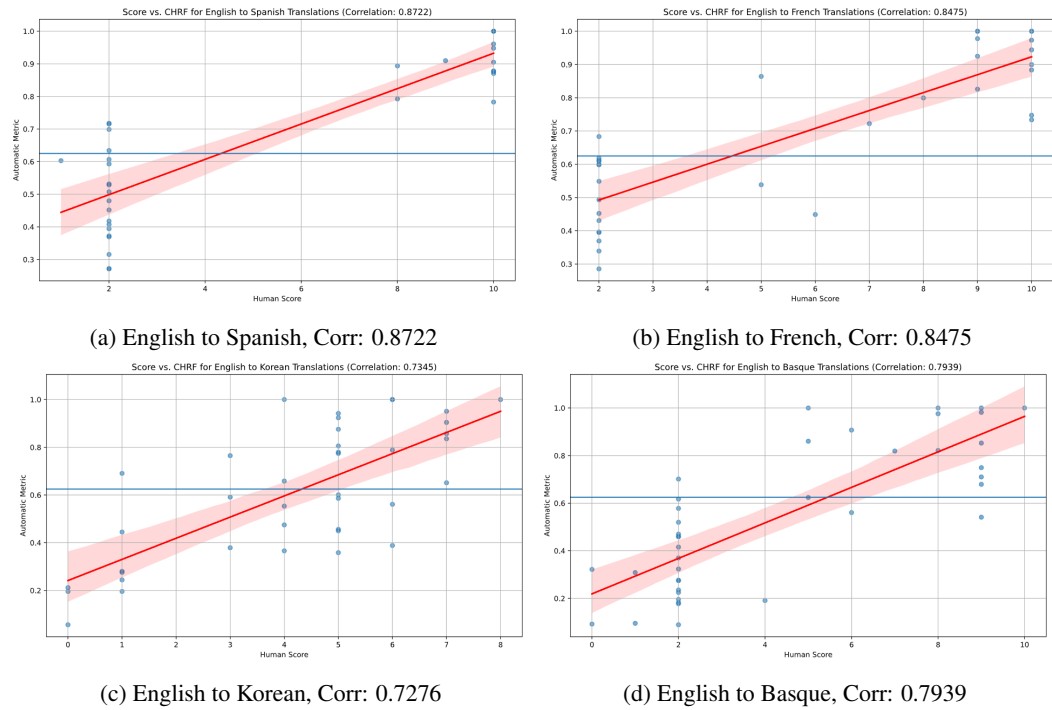

(a) English to Spanish, Corr: 0.8722        (b) English to French, Corr: 0.8475

(c) English to Korean, Corr: 0.7276        (d) English to Basque, Corr: 0.7939

Figure 11: Human Score vs. CHRF for manually evaluated translations. Strong correlation confirms reliability of the automatic metric.

## C  FURTHER INFORMATION ABOUT THE EXPERIMENTS

### C.1  MODEL CHOICES

We selected the models to benchmark M3Kang in order to cover both open and closed models, families of models of different sizes, as well as models that were specifically trained for reasoning (reasoning models) and other more general ones. The current list is depicted in Table 6, and we plan on extending it further. Importantly, the price range of closed models is below $ 5 for input tokens and $ 15 for output tokens, and open models fit inside an H100 GPU for inference. More detail on the cost of our experiments can be found in Appendix E.

Table 6: Summary of models used in our experiments.

| Short name | Parameters | Open-source? | Model name |
|---|---|---|---|
| Gemma-1B | 1.0B | ✔ | google/gemma-3-1b-it |
| Moondream | 1.9B | ✔ | vikhyatk/moondream2 |
| Qwen-2.5-3B | 3.8B | ✔ | Qwen/Qwen2.5-VL-3B-Instruct |
| Gemma-4B | 4.3B | ✔ | google/gemma-3-4b-it |
| Phi-4 | 5.6B | ✔ | microsoft/Phi-4-multimodal-instruct |
| Qwen-2.5-7B | 8.3B | ✔ | Qwen/Qwen2.5-VL-7B-Instruct |
| Gemma-12B | 12.2B | ✔ | google/gemma-3-12b-it |
| Kimi-Thinking-2506 | 16.4B | ✔ | moonshotai/Kimi-VL-A3B-Thinking-2506 |
| GPT-4o | ? | ✘ | GPT-4o |
| Claude-4-Sonnet | ? | ✘ | Claude-4-Sonnet |
| Gemini-2.5-Pro | ? | ✘ | Gemini-2.5-Pro |

## C.2 LANGUAGE CHOICES FOR THE BENCHMARKS

We selected 16 out of the 108 languages in M3Kang for the benchmarks based on two main criteria: varying levels of internet representation (to study its relationship with multilingual reasoning performance) and diversity across linguistic families or subfamilies, to ensure a rich evaluation. Although Chinese does not use the Latin script, this is not an issue because most standard multiple-choice tests still label options with letters such as A, B, C, D, and E. Table 7 reports the percentage of internet representation for each selected language and the corresponding classification into resource categories (high, mid, or low). Data on language representation on the internet were obtained from W3Techs - Web Technology Surveys (2025) and compiled in Table 7.

Table 7: Percentage of representation of each used language on the Internet. High-resource: web presence % over $\in [1, 100]$; Mid-resource: web presence % over $\in [0.1, 1)$; Low-resource: web presence % over $\in [0, 0.1)$.

| Language | Abbr. | Resources on the Internet (%) | Category |
|---|---|---|---|
| English | eng | $4.92 \times 10^1$ | High |
| Spanish | spa | $6.00 \times 10^0$ | High |
| German | deu | $5.90 \times 10^0$ | High |
| French | fra | $4.40 \times 10^0$ | High |
| Turkish | tur | $1.70 \times 10^0$ | High |
| Chinese | zho | $1.1 \times 10^0$ | High |
| Vietnamese | vie | $1.03 \times 10^0$ | High |
| Indonesian | ind | $9.82 \times 10^{-1}$ | Mid |
| Lithuanian | lit | $1.73 \times 10^{-1}$ | Mid |
| Catalan | cat | $1.02 \times 10^{-1}$ | Mid |
| Estonian | est | $1.01 \times 10^{-1}$ | Mid |
| Afrikaans | afr | $2.50 \times 10^{-3}$ | Low |
| Swahili | swh | $1.70 \times 10^{-3}$ | Low |
| Maltese | mlt | $4.30 \times 10^{-4}$ | Low |
| Lingala | lin | $1.60 \times 10^{-5}$ | Low |
| Tsonga | tso | $6.00 \times 10^{-6}$ | Low |

To obtain detailed figures for Vietnamese and below (in terms of representation), we used the more detailed view or a comparison against similarly behaving languages to give us an idea of a more precise number. The exact detailed views and comparisons that we used are: Vietnamese (detailed), Indonesian (detailed), Lithuanian (detailed), Catalan (detailed), Estonian (detailed), Afrikaans (detailed), Swahili (detailed), Maltese (detailed), Lingala compared with Cebuano, and Tsonga compared with Lezghian.

## C.3 INDEPENDENCE OF CORRECTLY TRANSLATED SUBSETS

When comparing a model's performance on M3Kang across a set of languages, in order to guarantee a completely fair comparison, one should evaluate it on the intersection of correctly translated samples across the set of languages. However, when doing so across a large set of languages, the dataset size can be greatly reduced and in the worst case, decrease exponentially[1]. To avoid this, we verify our hypothesis that the accuracy of the model does not depend on the subset of correctly translated samples, essentially ensuring that problem statement translation success (as defined by our metrics, see B.3) and problem difficulty are independent. To do this, we conduct a hypothesis test that the accuracy in English of models in a subset of well-translated samples for a language is the same than the accuracy computed with all samples. We report the results in the table below.

---

[1]This is when the mistranslated samples are independent, the cardinal of the intersection of a dataset of size $N$ across $L$ languages is $Np^L$ where $p$ is the probability that a sample is correctly translated to a particular language.

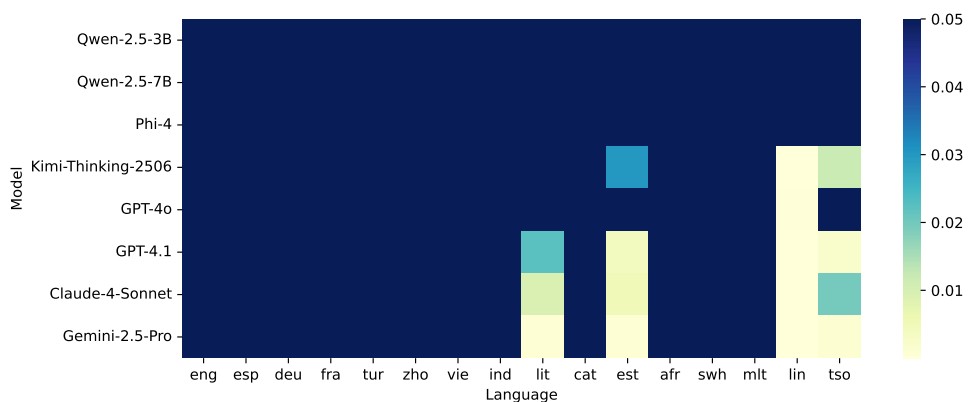

Figure 12: Heatmap of $p$-values for the test for equality between the accuracy computed with the correctly translates samples of a language and the accuracy computed with full set of samples.

We observe that in most cases, we cannot reject that the accuracies are the same at a 95% level, with the exception of Lingala, Tsonga, Estonian and Lithuanian which show statistically significant differences across the most capable models. We therefore opt to discard those four languages for evaluation, seeing as comparison with other subsets of the dataset might be biased. This gives us reason to believe that we are not significantly altering a model's performance when comparing accuracies from correctly translated samples of different languages.

### C.4 STEERING IN MULTIMODAL CONTEXTS

In this appendix, we present experiments conducted to determine how to achieve the best results when using the steering technique.

#### C.4.1 MULTILINGUAL STEERING CONFIGURATION

Previous work (Mahmoud et al., 2025; Zhang et al., 2025) has employed steering techniques similar to those described in Section 4.3 to enhance performance on multilingual datasets. However, because these studies neither provide detailed descriptions of their implementations nor share key experimental settings, we include in this appendix the results of our own experiments on MGSM (Shi et al., 2022a). The best configuration identified from these experiments is later applied to the M3Kang dataset. We chose MGSM for this exploratory phase to ensure that our findings are both generalizable to other contexts and comparable with results reported by prior work.

We evaluated the impact of five components when steering Qwen-3B:

1. **Which parts of the text to steer?** We compared steering *System + Question + Answer*, *System + Question*, and *Question only*.

2. **Vector source:** We compared vectors from the general-purpose FLORES dataset (NLLB Team et al., 2022) versus math-specific vectors from MGSM.

3. **Steering back:** We compared two-way steering (forward in early layers, backward in later layers) versus one-way steering (forward only, answering in English).

4. **Number of steered layers:** We compared steering 1 layer versus 5 layers, in both directions.

5. **Hyperparameter $c$:** We tested the values $1/N$, $2/(3N)$, and $1/(2N)$, where $N$ is the number of steered layers.

All possible combinations of these configurations were tested over three runs on the German subset of MGSM; we report only the mean for each factor, in Table 8.

From Table 8, we extract that **(1)** it is best to not steer the answer, and unclear whether to steer the system instructions or not, **(2)** it is better to use vectors from FLORES than MGSM, probably due

Table 8: Mean accuracies for different steering configurations on the German subset of MGSM.

| Factor | Option 1 | Option 2 | Option 3 |
|---|---|---|---|
| **Parts to steer** | Sys+Q+A - 58.3% | Sys+Q - *62.4%* | Q only - **62.5%** |
| **Vector source** | FLORES - **62.4%** | MGSM - 59.2% | – |
| **Steering back** | Two-way - **60.9%** | One-way - 24.9% | – |
| **Layers steered** | 1 layer - *60.1%* | 5 layers - **61.5%** | – |
| $c$ **hyperparameter** | $1/N$ - 58.4% | $2/(3N)$ - 61.0% | $1/(2N)$ - **62.5%** |

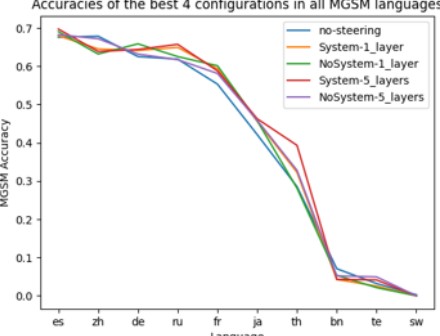 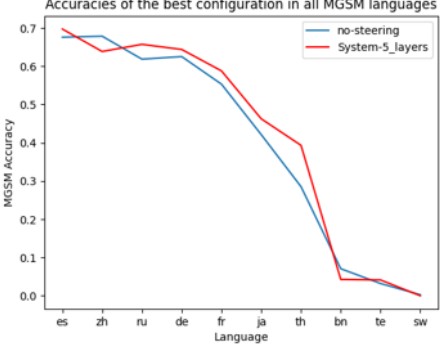

Figure 13: **Right:** Best 4 possible combinations of steering used in all languages of MGSM against no steering (vanilla). **Left:** Best combination overall compared against vanilla.

to its higher variety in the texts, **(3)** it is best to steer back and forward, **(4)** it is unclear whether it is best to steer one or five layers (we didn't test higher values since that would imply a big part of the network being steered), and **(5)** a $c$ value of around $1/(2N)$ is best.

Therefore, with these results we still have 4 possible combinations to use, which we run against the full MGSM dataset to find which works best. Using the results from Figure 13, we conclude to steer 5 layers, and steer the system instructions as well as the question. Finally, after further hyperparameter tuning, we found that among Qwen-3B's 36 layers, it is best to begin forward steering at layer 6 and backward steering at layer 21.

### C.4.2   QWEN-7B HYPERPARAMETERS

When attempting to use similar hyperparameters from Qwen-3B on Qwen-7B, we observed a significant drop in performance on the MGSM dataset. As a result, we conducted further hyperparameter tuning specific to this model and dataset. Our findings indicate that, out of the 28 layers in Qwen-7B, the best configuration is to steer forward only layer 5 and steer backward only layer 20, using a value of 0.3 for the hyperparameter $c$.

### C.4.3   OPTIONS TOWARDS MULTIMODALITY

Since there are very few multilingual, multimodal reasoning datasets, the question of how to transfer this technique to the multimodal setting remains largely underexplored. We use M3Kang and Qwen-3B to address this gap by answering two related questions: **(1)** Is it better to compute steering vectors with text-only datasets or with datasets containing both text and images? **(2)** Should steering be applied to the images as well, or only to the text? Figure 14 compares the four possible combinations against a non-steered baseline. For the two options in (1), we compute the steering vectors using the text-only dataset FLORES (NLLB Team et al., 2022) and the text+image dataset mOSCAR (Futeral et al., 2025).

Our experiment shows that steering images in addition to text decreases performance. Moreover, when images are present in the prompt, it is preferable to use steering vectors computed from

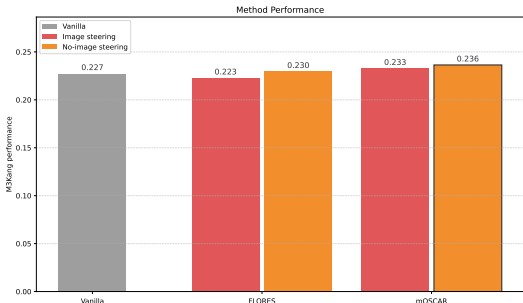

Figure 14: Performance of multimodal steering strategies. Text-only steering (orange) outperforms image+text steering (red), with mOSCAR achieving better results than FLORES when used to compute the steering vectors.

datasets containing both text and images, even if only the text is steered. Therefore, we adopted this configuration when benchmarking M3Kang using the steering vectors technique.

### C.5 FURTHER HUMANS VS VLMS EXPERIMENTS

In this appendix, we present additional experiments related to Section 4. While these results may not stand out individually, we believe they contribute meaningfully to the overall picture. The content is organized into four subsections:

1. Tables similar to Table 3 for the remaining high-performing models.

2. Distribution of text/image problems across levels.

3. Plots similar to Figure 8, covering all problems and text-only problems.

4. A table of advanced performance metrics.

#### C.5.1 PERCENTILE TABLE OF ALL MODELS

In the Kangaroo competition, students are encouraged to leave a question blank if they don't know the answer, as incorrect responses are penalized. This contrasts with how we evaluate VLMs, which are prompted to always select one of the five options. To fairly assess participant performance, we calculated their score as the number of correct answers plus one-fifth of the blank responses—representing the expected accuracy if they had guessed randomly. This allows for a fair comparison between human performance and that of the VLMs. Rather than comparing raw accuracy alone, we evaluate how VLMs would perform on the actual 2024 test relative to human participants, normalized by the number of questions per level (which is fewer than the total due to our filtering criteria). All VLM performance is reported in English. The results are quite revealing and can be found in Table 9. In summary, we already know that Gemini is the top-performing VLM, achieving a solid average of 76.4% in English according to Table 2, however, while it (almost) consistently ranks above 99% of the participants, even for low-resource languages, it never takes the top spot against participants.

#### C.5.2 AMOUNT OF TEXT/IMAGE PROBLEMS PER LEVEL

As discussed in Section 4.4, there is a clear performance gap between figure-based and text-only problems for VLMs. This means that depending on how the dataset is partitioned, some subsets may not offer a fully fair basis for comparison. A notable example is the dataset segmented by difficulty levels. As illustrated in Figure 15, lower levels contain a higher proportion of problems involving figures. Interestingly, when considering the dataset as a whole, approximately half of the problems include figures, while the other half do not.

Table 9: Percentile ranks (↑) that each model would have achieved on the 2024 test, segmented by problem difficulty level and language classes based on Internet presence.

| Percentile ranks (↑) | Lvl 0 | Lvl 1 | Lvl 2 | Lvl 3 | Lvl 4 | Lvl 5 | Lvl 6 | Lvl 7 |
|---|---|---|---|---|---|---|---|---|
| **Gemini-2.5-Pro** | | | | | | | | |
| English | 86.7% | 95.7% | 97.6% | 99.7% | 99.8% | 99.9% | 99.9% | 99.6% |
| High-resource langs. | 86.7% | 95.7% | 97.8% | 99.7% | 99.8% | 99.8% | 99.9% | 99.5% |
| Mid-resource langs. | 91.2% | 97.8% | 98.1% | 99.9% | 97.5% | 99.8% | 99.9% | 99.6% |
| Low-resource langs. | 88.3% | 95.6% | 97.6% | 99.7% | 99.8% | 99.8% | 99.9% | 99.4% |
| **Claude-4-Sonnet** | | | | | | | | |
| English | 72.4% | 86.0% | 93.6% | 99.5% | 99.4% | 99.4% | 99.8% | 98.3% |
| High-resource langs. | 72.4% | 87.6% | 92.4% | 99.1% | 99.0% | 98.8% | 99.4% | 96.2% |
| Mid-resource langs. | 72.4% | 88.6% | 93.6% | 99.1% | 99.0% | 99.0% | 99.1% | 95.6% |
| Low-resource langs. | 63.9% | 76.2% | 91.6% | 97.8% | 98.2% | 97.5% | 98.6% | 92.3% |
| **GPT-4o** | | | | | | | | |
| English | 36.8% | 47.3% | 61.8% | 93.6% | 89.4% | 89.3% | 77.0% | 61.8% |
| High-resource langs. | 35.3% | 51.5% | 54.5% | 91.3% | 86.4% | 82.5% | 74.5% | 50.4% |
| Mid-resource langs. | 44.4% | 51.5% | 54.5% | 91.3% | 89.4% | 82.5% | 78.9% | 64.0% |
| Low-resource langs. | 33.8% | 47.3% | 39.0% | 84.1% | 70.5% | 67.8% | 60.4% | 41.7% |
| **Kimi-Thinking-2506** | | | | | | | | |
| English | 47.3% | 57.5% | 73.8% | 96.9% | 95.3% | 94.5% | 95.4% | 78.1% |
| High-resource langs. | 36.8% | 56.3% | 64.4% | 93.6% | 91.2% | 89.3% | 88.1% | 66.2% |
| Mid-resource langs. | 36.8% | 57.5% | 67.9% | 93.6% | 91.2% | 90.6% | 88.1% | 67.7% |
| Low-resource langs. | 11.0% | 14.5% | 19.2% | 41.5% | 33.7% | 24.0% | 18.0% | 21.2% |
| **Qwen-2.5-7B** | | | | | | | | |
| English | 33.8% | 32.2% | 54.5% | 85.5% | 85.0% | 75.2% | 72.0% | 55.1% |
| High-resource langs. | 20.1% | 25.7% | 25.9% | 61.1% | 66.9% | 55.0% | 40.7% | 27.9% |
| Mid-resource langs. | 22.4% | 32.2% | 19.2% | 52.4% | 52.9% | 55.0% | 34.2% | 21.2% |
| Low-resource langs. | 8.0% | 9.8% | 13.7% | 20.3% | 28.0% | 19.7% | 11.8% | 11.2% |
| **Phi-4** | | | | | | | | |
| English | 18.6% | 29.4% | 25.9% | 55.8% | 61.7% | 39.7% | 22.3% | 17.3% |
| High-resource langs. | 9.1% | 11.8% | 23.1% | 36.3% | 31.1% | 24.0% | 16.0% | 13.5% |
| Mid-resource langs. | 10.1% | 15.3% | 23.1% | 36.3% | 18.0% | 19.7% | 11.8% | 14.8% |
| Low-resource langs. | 8.0% | 14.5% | 19.2% | 24.8% | 14.4% | 17.4% | 11.8% | 8.5% |
| **Qwen-2.5-3B** | | | | | | | | |
| English | 20.1% | 25.7% | 34.9% | 63.4% | 55.4% | 45.0% | 43.9% | 17.3% |
| High-resource langs. | 18.6% | 13.4% | 21.3% | 41.5% | 35.8% | 33.2% | 22.3% | 13.5% |
| Mid-resource langs. | 18.6% | 29.4% | 23.1% | 36.3% | 35.8% | 22.0% | 20.2% | 16.0% |
| Low-resource langs. | 4.5% | 3.7% | 11.8% | 22.5% | 14.4% | 12.1% | 7.7% | 8.5% |

### C.5.3 ACCURACY GROUPED BAR-CHARTS BY LEVELS

Here we present the equivalent plots to those in Figure 8, but instead of focusing on the subset of figure-based problems, we consider either the subset of text-only problems or the entire dataset. As expected, for text-only problems, the performance of VLMs tends to decline as the problem level increases. However, when considering the full dataset, the overall trend is reversed: similar to the figure-based subset, accuracy improves with higher level problems.

### C.5.4 ADVANCED HUMAN VS VLMS METRICS

In this subsection, we aim to replicate the results presented by Cherian et al. (2024). Specifically, we reproduce the following metrics, which are described in greater detail in the original paper:

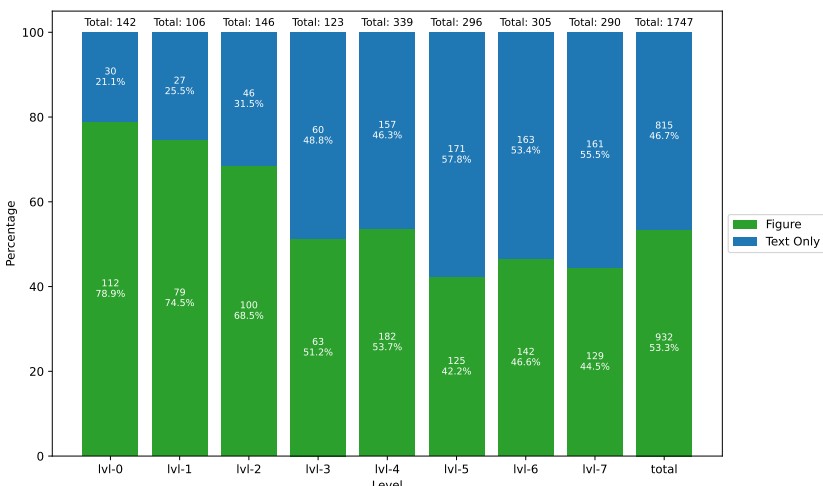

Figure 15: Percentage of the amount of figure-based problems and text-only problems in M2Kang (the English subset of M3Kang) across levels and total.

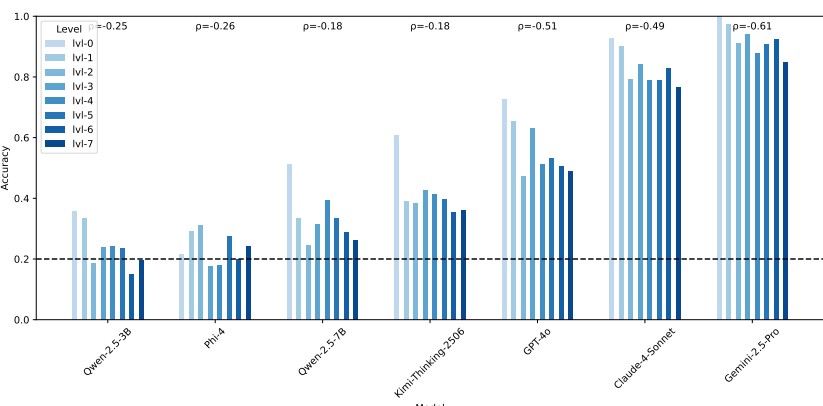

Figure 16: Accuracy of problems that do not contain figures across levels and models, with corresponding Spearman's correlation coefficients (accuracy vs level).

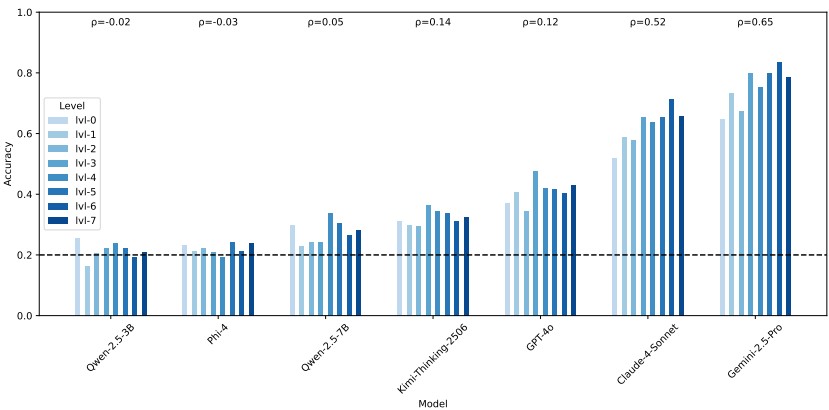

Figure 17: Overall accuracy across levels and models, with corresponding Spearman's correlation coefficients (accuracy vs level).

**Difficulty Index:**   This metric represents the average accuracy of all participants for a given problem. We report the Spearman's rank correlation coefficient between this index and the accuracy of VLMs.

**Difficulty Index 1% (new):**   This is the average accuracy of the top 1% of participants for each problem. We report the Spearman's rank correlation coefficient between this metric and VLM accuracy.

**Discriminative Index:**   Defined as the difference in average accuracy between the top 20% and bottom 20% of participants. We report the Spearman's rank correlation coefficient between this index and VLM accuracy.

**Weight Correlation:**   For each level, problems are further divided into three difficulty blocks by the organizers: easy, medium, and hard. We assign weights of 0.33, 0.66, and 1.0 to problems in these blocks, respectively, and report the Spearman's rank correlation coefficient between these weights and VLM accuracy.

The results of this analysis did not align with those reported by Cherian et al. (2024). As shown in Table 10, we were unable to reproduce the same patterns. While some trends do emerge—for instance, levels 2 and 3 show a more negative correlation with human performance, whereas levels 1 and 7 exhibit a more positive correlation—we were unable to identify a clear explanation for these findings. This discrepancy suggests that further investigation is needed to understand the underlying factors influencing the relationship between problem difficulty and VLM performance.

Table 10: Advanced metrics of human against VLMs performance

| Metric | Model | Lvl-0 | Lvl-1 | Lvl-2 | Lvl-3 | Lvl-4 | Lvl-5 | Lvl-6 | Lvl-7 |
|---|---|---|---|---|---|---|---|---|---|
| Diff-I | Gemini-2.5-Pro | -0.51 | -0.20 | -0.27 | -0.54 | 0.07 | 0.13 | 0.44 | -0.09 |
| | Claude-4-Sonnet | -0.39 | 0.92 | 0.07 | -0.44 | 0.19 | 0.23 | 0.03 | 0.05 |
| | GPT-4o | 0.31 | 0.38 | 0.02 | -0.54 | 0.14 | -0.05 | -0.37 | 0.48 |
| | Kimi-Thinking-2506 | -0.01 | 0.17 | -0.08 | -0.57 | -0.19 | 0.10 | 0.01 | 0.23 |
| | Qwen-2.5-7B | 0.18 | 0.18 | 0.10 | 0.05 | -0.27 | -0.45 | 0.10 | 0.05 |
| | Phi-4 | 0.30 | 0.09 | -0.24 | 0.17 | -0.30 | -0.41 | 0.32 | 0.08 |
| | Qwen-2.5-3B | 0.37 | 0.13 | -0.04 | -0.31 | 0.06 | -0.14 | 0.04 | 0.11 |
| Diff-I-1% | Gemini-2.5-Pro | -0.56 | -0.37 | -0.36 | -0.58 | 0.31 | 0.32 | 0.26 | 0.26 |
| | Claude-4-Sonnet | -0.54 | 0.82 | -0.20 | -0.45 | 0.32 | 0.31 | -0.24 | 0.26 |
| | GPT-4o | 0.21 | 0.21 | -0.18 | -0.39 | -0.05 | 0.08 | 0.00 | 0.60 |
| | Kimi-Thinking-2506 | -0.15 | 0.10 | -0.25 | -0.56 | 0.01 | 0.05 | 0.10 | 0.28 |
| | Qwen-2.5-7B | 0.12 | 0.15 | 0.20 | 0.00 | -0.29 | -0.37 | -0.05 | 0.35 |
| | Phi-4 | 0.20 | 0.35 | -0.45 | 0.07 | -0.16 | -0.47 | 0.41 | 0.04 |
| | Qwen-2.5-3B | 0.61 | 0.16 | 0.04 | -0.15 | 0.37 | -0.18 | -0.07 | 0.07 |
| Disc-I | Gemini-2.5-Pro | 0.36 | -0.05 | -0.38 | -0.40 | 0.31 | 0.40 | 0.32 | 0.09 |
| | Claude-4-Sonnet | 0.11 | 0.47 | -0.09 | -0.29 | 0.39 | 0.33 | -0.13 | 0.24 |
| | GPT-4o | 0.43 | 0.38 | -0.15 | -0.53 | -0.02 | 0.14 | -0.23 | 0.63 |
| | Kimi-Thinking-2506 | 0.21 | 0.53 | -0.13 | -0.53 | 0.01 | 0.12 | -0.06 | 0.28 |
| | Qwen-2.5-7B | -0.12 | 0.68 | 0.30 | -0.03 | -0.22 | -0.30 | 0.13 | 0.19 |
| | Phi-4 | 0.00 | 0.17 | -0.24 | 0.12 | -0.18 | -0.38 | 0.50 | -0.21 |
| | Qwen-2.5-3B | 0.34 | -0.03 | 0.16 | -0.20 | 0.24 | -0.05 | -0.03 | 0.03 |
| Weight-I | Gemini-2.5-Pro | 0.43 | 0.00 | 0.28 | 0.35 | -0.25 | -0.13 | -0.24 | 0.04 |
| | Claude-4-Sonnet | 0.41 | -0.61 | -0.04 | 0.20 | -0.28 | -0.17 | 0.12 | -0.10 |
| | GPT-4o | -0.28 | -0.72 | 0.12 | 0.27 | -0.35 | -0.04 | 0.24 | -0.43 |
| | Kimi-Thinking-2506 | 0.02 | -0.52 | 0.09 | 0.17 | 0.06 | 0.08 | 0.31 | -0.13 |
| | Qwen-2.5-7B | -0.04 | -0.22 | -0.28 | -0.01 | 0.10 | 0.29 | -0.21 | -0.40 |
| | Phi-4 | -0.33 | 0.19 | 0.09 | -0.05 | 0.04 | 0.44 | -0.48 | -0.17 |
| | Qwen-2.5-3B | -0.51 | -0.09 | -0.02 | -0.02 | -0.07 | 0.12 | 0.05 | 0.20 |

## C.6 ANALYSIS OF PROBLEM CATEGORY AND MODALITY

In this appendix, we present an analysis of the impact on performance of problem category and the reasoning components that it might require, complementing the one in Section 4.4. We have classified the English version of the problems into the following five classes: Algebra, Arithmetic, Combinatorics & Probability, Geometry, and Logic.

To perform this classification, we used a VLM, with the prompt labeled *System prompt for problem classification* in Appendix D. This allowed us to accurately categorize problems that included diagrams or other visual elements, ensuring consistency across modalities. To validate the reliability of this automated classification, we sampled a subset of 48 problems (6 for each level) and compared the VLM's assignments with human classifications. The results showed strong agreement, confirming that the VLM-based approach was robust and aligned with expert judgment. This classification covers all problems in the dataset, since they all come from a common pool, which English covers completely. In Table 11, the amount of problems per category separated by figure-based/text-only is shown.

Table 11: Problem count in each class.

| Category | Figure | Text-Only | Total |
|---|---|---|---|
| Algebra | 24 | 154 | 178 |
| Arithmetic | 35 | 199 | 234 |
| Combinatorics & Probability | 83 | 153 | 236 |
| Geometry | 513 | 90 | 603 |
| Logic | 277 | 219 | 496 |
| **Total** | 932 | 815 | 1747 |

In Figure 18, we show the accuracy of the VLMs with which we have been benchmarking throughout the paper separated by problem category. Figure 19 shows the same information separated into two plots, one for each text-based and one for figure-based problems. We observe that the models perform better in Algebra and Arithmetic problems, and that they perform consistently worse in Logic and Geometry. Combinatorics & Probability is mixed, with performance closer to Geometry and Combinatorics & Probability for smaller models, and performance closer to Algebra and Arithmetic in larger/closed-source models.

Focusing on both classifications at the same time, we observe in Table 11 that Geometry and Logic have a high presence of figure-based problems, so the low performance on these two fields could be attributed to the high presence of figure-based problems. Further confirmation for this comes from observing Figure 19, where we can see a significant difference in performance in Geometry and Logic when needing or not needing to use a figure to solve a problem. This could be signaling a lower performance when using image tokens compared to when not needing them for a problem, highlighting possible performance improvement avenues for VLMs.

In a more individualized problem category analysis, we observe that generally the performance trend is:

$$\text{Logic} \sim \text{Geometry} < \text{Combinatorics \& Probability} < \text{Arithmetic} < \text{Algebra}$$

While the authors of this paper lack the tools to understand the deeper reasons behind this difference in performance when comparing areas of mathematical reasoning, we believe that this might be an interesting avenue to explore for deeper insights about how VLMs (or LLMs) work with abstractions.

## D PROMPTS

This appendix contains the complete set of prompts employed throughout the project. We include three categories of prompts: (i) those used as instructions for interacting with and utilizing the dataset, (ii) those used in the *LLM-as-a-judge* settings for the dataset generation and (iii) those used for classifying problems into categories. By providing the full prompt specifications, we aim to ensure transparency and reproducibility of our experimental setup.

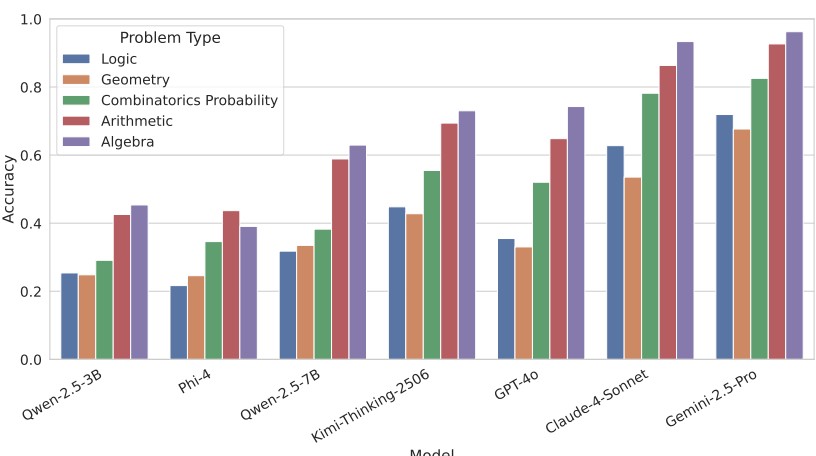

Figure 18: Overall accuracy of each model in English, separated by problem category.

For the experiments in Section 4, we used the following system prompts for every one of the 15 languages tested. For each problem, the user prompt consisted of the problem image along with its text version.

| English |
| --- |
| Analyze the question shown in the image and given in the text, and choose the correct answer from the options provided.

**Instructions**: Explain your reasoning and provide your final answer in this specific format, without changes:

Reasoning: Describe the thought process that led you to the answer.
Answer: A), B), C), D) or E) |

| Catalan |
| --- |
| Analitzeu la pregunta que apareix a la imatge i en el text, i escull la resposta correcta de les opcions proporcionades.

**Instruccions**: Explica el teu raonament i proporciona la teva resposta final en aquest format específic, sense canvis:

Raonament: Descriu el procés de pensament que t'ha portat a la resposta.
Resposta: A), B), C), D) o E) |

| Afrikaans |
| --- |
| Ontleed die vraag wat in die prent en in die teks getoon word, en kies die korrekte antwoord uit die opsies wat voorsien word.

**Instruksies**: Verduidelik jou redenasie en verskaf jou finale antwoord in hierdie spesifieke formaat, sonder veranderinge:

Redenasie: Beskryf die denkproses wat jou tot die antwoord gelei het.
Antwoord: A), B), C), D) of E) |

| German |
| --- |
| Analysieren Sie die im Bild und im Text dargestellte Frage und wählen Sie die richtige Antwort aus den zur Verfügung gestellten Optionen.

**Anleitung**: Erklären Sie Ihre Argumentation und geben Sie Ihre endgültige Antwort in diesem spezifischen Format ohne Änderungen an:

Begründung: Beschreiben Sie den Denkprozess, der zu Ihrer Antwort geführt hat.
Antwort: A), B), C), D) oder E) |

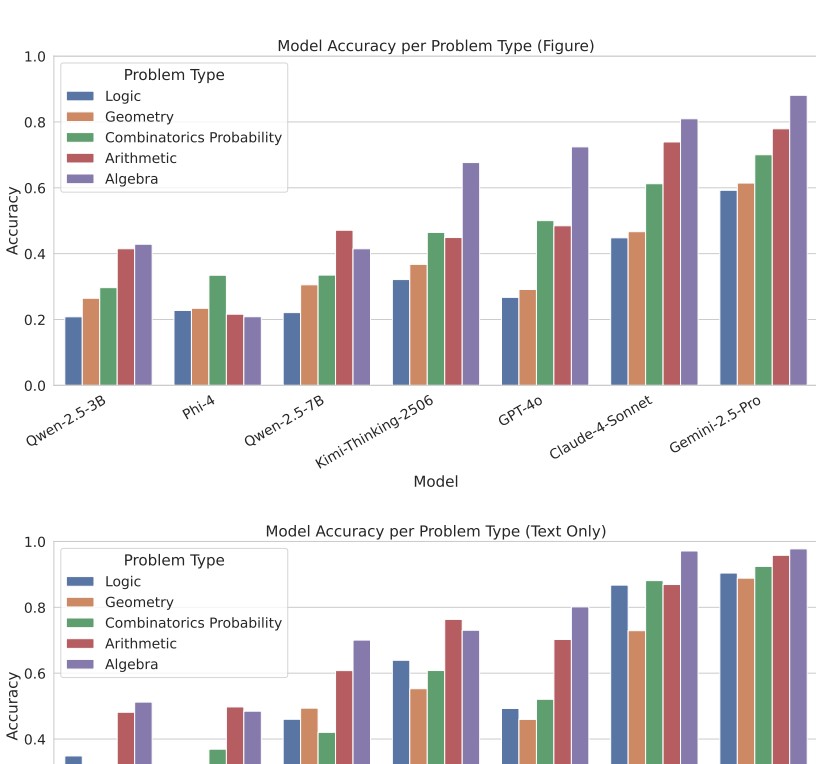

Figure 19: Accuracy of each model in English, separated by problem category and figure-based/text-only.



**Spanish**

Analiza la pregunta que aparece en la imagen y en el texto, y elige la respuesta correcta de las opciones proporcionadas.

**Instrucciones**: Explica tu razonamiento y proporciona tu respuesta final en este formato específico, sin cambios:

Razonamiento: Describe el proceso de pensamiento que te llevó a la respuesta.
Respuesta: A), B), C), D) or E)





**French**

Analysez la question montrée dans l'image et donnée dans le texte, et choisissez la bonne réponse parmi les options fournies.

**Instructions**: Expliquez votre raisonnement et fournissez votre réponse finale dans ce format spécifique, sans changements:

Raisonnement: Décrivez le processus de pensée qui vous a conduit à la réponse.
Réponse: A), B), C), D) ou E)



### Indonesian

Analisis pertanyaan yang ditunjukkan dalam gambar dan yang diberikan dalam teks, dan pilih jawaban yang benar dari pilihan yang disediakan.

**Petunjuk**: Jelaskan alasan Anda dan berikan jawaban akhir Anda dalam format khusus ini, tanpa perubahan:

Alasan: Jelaskan proses berpikir yang membawa Anda ke jawaban.
Jawaban: A), B), C), D) atau E)

### Maltese

Analizza l-mistoqsija murija fl-immaġni u mogtija fit-test, u agżel it-tweġiba t-tajba mill-gażliet ipprovduti.

**Istruzzjonijiet**: Spjega r-raġunament tiegek u agti t-tweġiba finali tiegek f'dan il-format speċifiku, mingajr tibdil:

Raġunament: Iddeskrivi l-proċess ta' sieb li wassallek gat-tweġiba.
Tweġiba: A), B), C), D) jew E)

### Swahili

Kuchambua swali inavyoonekana katika picha na aliyopewa katika maandishi, na kuchagua jibu sahihi kutoka chaguzi zinazotolewa.

**Maagizo**: Eleza hoja yako na kutoa jibu lako la mwisho katika muundo huu maalum, bila mabadiliko:

Hoja: Eleza mchakato wa mawazo ambayo imesababisha wewe jibu.
Jibu: A), B), C), D) au E)

### Turkish

Resimde gösterilen ve metinde verilen soruyu analiz edin ve verilen seçeneklerden doğru cevabı seçin.

**Talimatlar**: Düşüncenizi açıklayın ve son cevabınızı bu özel biçimde, değişiklik yapmadan verin:

Gerekçe: Cevaba ulaşmanızı sağlayan düşünme sürecini açıklayın.
Cevap: A), B), C), D) veya E)

### Lithuanian

Išnagrinėkite paveikslėlyje ir tekste pateikta klausima ir pasirinkite teisinga atsakyma iš pateiktu variantu.

**Instrukcijos**: Paaiškinkite savo argumentus ir pateikite galutini atsakyma šiuo konkrečiu formatu, be pakeitimu:

Argumentavimas: Apibūdinkite mastymo procesa, kuris paskatino jus atsakyti.
Atsakymas: A), B), C), D) arba E)

### Estonian

Analüüsige pildil ja tekstis esitatud küsimust ning valige esitatud valikute hulgast õige vastus.

**Juhised**: Selgita oma põhjendusi ja anna oma lõplik vastus selles konkreetses vormingus, ilma muudatusteta:

Arvestamine: Kirjelda vastuse leidmiseni viinud mõtteprotsessi.
Vastus: A), B), C), D) või E)

### Lingala

Talela motuna oyo ezali na elilingi mpe oyo ezali na makomi, mpe poná eyano ya malamu na kati ya biyano oyo ezali.

**Mibeko**: Limbolá makanisi na yo mpe pesá eyano na yo ya nsuka na ndenge oyo, kozanga kobongola yango:

Makanisi: Lobelá ndenge oyo okanisaki liboso opesa eyano.
Eyano: A), B), C), D) to E)

### Tsonga

Hlamusela xivutiso lexi kombisiweke eka xifaniso na lexi nga tsariwa eka tsalwa, kutani u hlawula nhlamulo leyi faneleke eka leti nga kona.

**Swileriso**: Hlamusela ndlela leyi u ehleketaka ha yona kutani u nyikela nhlamulo ya wena yo hetelela hi ndlela leyi, handle ko cinca:

Ku ehleketisisa: Hlamusela ndlela leyi u ehleketeke ha yona leyi endleke leswaku u kuma nhlamulo.
Nhlamulo: A), B), C), D) kumbe E)

### Vietnamese

Phân tích câu hỏi trong hình và trong văn bản, và chọn câu trả lời đúng trong các lựa chọn được cung cấp.

**Hướng dẫn**: Giải thích lý luận của bạn và cung cấp câu trả lời cuối cùng của bạn trong định dạng cụ thể này, không thay đổi:

Lý do: Mô tả quá trình suy nghĩ dẫn bạn đến câu trả lời.
Câu trả lời: A), B), C), D) hoặc E)

Below is the prompt used in Sections 3.1 and 3.2 for automatic corruption generation. As described in those sections, this prompt served to flag potentially corrupted samples in the dataset pipeline, which were subsequently reviewed manually.

### Corruption detection

You are given a math question in two formats:

- An image with the question, figures, and answer choices.
- A natural language transcription.

Classify the question as:
- 'corrupted' — if it includes the answer, has parsing artifacts (e.g., garbled text, formatting issues), hides key visual info, or if image and text don't match.
- 'not corrupted' — if the question is complete, consistent, and clear.

Return ONLY the label.

Finally, below are the prompts used for the problem classification from Appendix C.6 and for the figure versus text-only analysis in Section 4.4.

### System prompt for problem classification

You are an expert math problem classifier. You classify a given problem, possibly accompanied by a figure, into one of these classes: geometry, algebra, arithmetic, combinatorics-probability, logic. You only reply with the name of the class.

> **System prompt for figure detection**
>
> You are an image classifier that determines whether an image contains a figure, or is text-only. You only reply with either "figure" or "text-only".

# E    COST ESTIMATION

The total cost required for this project can be divided into (1) GPU-hours used by the open-source models and (2) the API calls made by the closed-source models.

Using an estimated cost of 2 USD per GPU-hour (this takes into account amortization costs, maintenance and energy cost), we summarize the total cost for open models in the table below:

Table 12: Summary of costs related to open-source models.

| Cost Category | Cost (USD) |
|---|---|
| Dataset Creation | 191.88 |
| Backtranslation | 128.59 |
| Vanilla (open models) | 3316.58 |
| MTR | 864.77 |
| CLSP | 3459.09 |
| Steering | 241.88 |
| **Total (Open Models)** | **8202.79** |

Running the closed models on all 108 languages cost **2,375.25 USD** in API calls, bringing the total cost to about **10,578.04 USD**.

# F    USE OF LARGE LANGUAGE MODELS (LLMS)

Throughout the development of this work, we have employed Large Language Models (LLMs) to support some aspects of the writing and research process. Specifically, LLMs were utilized to:

- Detect and correct grammatical and stylistic errors.
- Refine and clarify written content for improved readability and coherence.
- Retrieve and summarize relevant academic literature to inform our analysis.

All outputs generated by LLMs were critically reviewed and edited to ensure accuracy, relevance, and alignment with the objectives of this work.

