# OpenReview forum: "M3Kang: Evaluating Multilingual Multimodal Mathematical Reasoning in Vision-Language Models"
_ICLR.cc/2026/Conference — Submitted to ICLR 2026_

### Official Review · Reviewer_yh29 · 2025-10-23

**Soundness:** 3
**Presentation:** 2
**Contribution:** 3
**Rating:** 6
**Confidence:** 2

**Summary:**

This paper proposes M3Kang, the first highly multilingual, graphically-based mathematical reasoning assessment dataset. This dataset draws 1,789 questions from the Kangaroo competition and expands to 15 languages, supported by an open automated translation and quality control pipeline. Benchmark results show that models significantly underperform text-only problems on questions containing diagrams. Cross-lingual performance is strongly correlated with internet coverage (more pronounced for smaller models). A simple "text + English translation parallel prompt" (MTR) approach is most effective in multimodal scenarios, significantly narrowing language gaps.

**Strengths:**

1. This paper introduces a multilingual, multimodal math reasoning benchmark across 15 languages (with an English subset), enabling fair cross-lingual comparison on a unified problem pool.
2. This paper provides an open, rigorous translation and quality-control pipeline (reference-free backtranslation metrics, LLM-as-judge) that preserves layout and is easily reusable.
3. This paper delivers comprehensive evaluations of open/closed VLMs and multilingual techniques, revealing a strong visual reasoning gap and establishing MTR as the most effective method; it also includes human comparisons with 68k students.

**Weaknesses:**

1. Dataset and code links are inaccessible
2. The dataset's statistical metrics are unclear. How many multimodal questions are there and how many are plain text?
3. Can you provide a comparison with existing multilingual and multimodal datasets, perhaps in a table format?
4. Regarding **sec. 4.4 Text-Only vs. FIGURE PROBLEMS**. The article observes that the accuracy rate for plain text questions is higher than for multimodal questions. Could this be because multimodal questions are inherently more difficult? I suggest adding a "comparative experiment at the same difficulty level" to make the argument more convincing.

**Questions:**

see weaknesses

---

> ### Author Response · Authors · 2025-11-21
> **Rebuttal by Authors**
>
> We thank the reviewer for their careful reading and helpful suggestions. Below we address each point raised, clarifying them and, where necessary, explaining how they have been resolved in the revised version.
>
> - *Dataset and code links are inaccessible.*
>
> Due to the double-blind review process, we cannot share the Hugging Face or GitLab repository links at this stage. These will be provided **immediately after the double-blind period concludes**.
>
> - *The dataset's statistical metrics are unclear. How many multimodal questions are there and how many are plain text?*
>
> Figure 15 in Appendix C.5 illustrates this distribution: **46.7% of problems are text-only, while 53.3% include a diagram**. It also breaks these percentages down by level, showing that the proportions remain fairly consistent overall, though lower levels tend to include more figures.
>
> - *Can you provide a comparison with existing multilingual and multimodal datasets, perhaps in a table format?*
>
> We have added a Table (see **Table 1 in Section 2**) comparing our dataset with existing datasets that contain multilingual, multimodal and mathematical questions. The table shows that our dataset includes by far the most languages (we cover 108 languages, more than two times the second best) and is the only one dedicated exclusively to mathematical multimodal reasoning, underscoring its value as a contribution.
>
> - *Regarding sec. 4.4 Text-Only vs. FIGURE PROBLEMS. The article observes that the accuracy rate for plain text questions is higher than for multimodal questions. Could this be because multimodal questions are inherently more difficult? I suggest adding a "comparative experiment at the same difficulty level" to make the argument more convincing.*
>
> We agree that adding a comparative experiment would strengthen the argument. To test whether multimodal questions are inherently more difficult, we **extended our Figure vs. Text-only experimental analysis** by comparing VLM accuracy within difficulty levels and, separately, comparing with human performance data (Figure 7 in the paper). In the former experiment, we see that VLM performs consistently better in text-only problems when compared with problems that contain figures within every level. In the latter experiment, we clearly see that problems that contain figures are clearly easier for humans, in contrast to VLMs, which find easier problems that are text-only. This reinforces our claim that multimodal questions are significantly harder for VLMs than text-only counterparts of similar difficulty, and that the difference is not attributable to easier problems. See Section 4.4 for further details.
>
> We trust these answers are satisfactory and respectfully encourage the reviewer to consider updating the score.

---

> > ### Comment · Reviewer_yh29 · 2025-11-24
> > **Reply to the authors**
> >
> > Thanks for your responses, it has addressed my previous concerns.
> >
> > Now I have a new question, I noticed that the authors expanded the language coverage from 15 to 108 languages, this progress relies exclusively on automated back-translation filtering. While back-translation can filter out obviously poor translations, such automatic metrics do not reliably capture deeper semantic distortions (subtle logical changes, symbol misinterpretation, mathematical ambiguities, or culturally-loaded numerical phrasing). This is particularly critical in mathematical reasoning datasets where small semantic mismatches can change the entire problem structure or its solvability.
> >
> > Did the authors conduct any human spot-checking or semantic audits for the multilingual versions, especially for low-resource languages where back-translation systems tend to be less reliable? If not, how do the authors ensure that incorrect or semantically drifted translations are not contributing to the observed performance gaps across languages?
> >
> > Some clarification on this would strengthen the validity claims for the multilingual evaluation results.

---

> > > ### Author Response · Authors · 2025-11-27
> > > **Reply to the reviewer**
> > >
> > > The reviewer is correct that the initial human audit only included high-resource languages (French and Spanish), which could introduce bias into the analysis. We have now extended the audit to cover one mid-resource language (Korean) and one low-resource language (Basque). The results are consistent with the previous findings, as the correlations between the automatic metric and the human metric remain high: 0.727 for Korean and 0.793 for Basque. Full details are provided in Appendix B.4.
> > >
> > > Addressing the specific concerns that the reviewer raised, we have the following points:
> > >
> > > - Potential symbol misinterpretation: During the initial filtering, we removed all complex symbols from the statements due to parsing issues, so this is not a source of error.
> > >
> > > - Culturally loaded numerical phrasing: We manually reviewed the entire Catalan dataset and its English translations and found no cultural ambiguities.
> > >
> > > - Subtle logical changes or mathematical ambiguities: This aspect is more difficult to control, which is why we performed a human audit on a subset of languages. The audit reveals a strong correlation between human and automatic metrics, suggesting that its overall impact is small.
> > >
> > > Another indication supporting that deep but subtle semantic distortions are not contributing to the observed performance gaps across languages is that Gemini 2.5 Pro (the best-performing model in our analysis) shows consistently stable performance across languages. This suggests that the potential impact of poor translation quality is minimal.
> > >
> > > Overall, while we acknowledge the possibility of some translation issues, all evidence suggests that their impact is small.

---

### Official Review · Reviewer_HBk9 · 2025-10-25

**Soundness:** 2
**Presentation:** 3
**Contribution:** 1
**Rating:** 2
**Confidence:** 4

**Summary:**

Build a highly multilingual, multimodal benchmark (M3Kang) from Kangaroo Math problems; create an automated, quality-controlled translation pipeline; benchmark SOTA VLMs; test multilingual inference techniques in multimodal settings; compare to human performance.

**Strengths:**

- A multilingual and multimodal math benchmark with a reproducible pipeline.
- This work offers a scalable data translation pipeline.
- Comprehensive benchmarking across open and closed models.

**Weaknesses:**

- Reliance on backtranslation may systematically disadvantage low-resource languages
- Cross-language fairness relies on filtered subsets, limited statistical testing of comparability.
- Some models (Gemma) perform below chance; analysis of why (prompting, vision adapters) is shallow.

**Questions:**

- Provide exact prompts per language and the LLM-as-judge criteria and models used.
- Clarify licensing and permissions for dataset redistribution.
- Why not include Chinese?

**Details Of Ethics Concerns:**

- Potential bias amplification across languages; risk of overclaiming language parity.
- Licensing/consent: clarify rights to redistribute problem statements and images from Kangaroo materials.

---

> ### Author Response · Authors · 2025-11-21
> **Rebuttal by Authors**
>
> We thank the reviewer for the time spent on our submission and for the constructive remarks. Below we address each point raised, clarifying them and, where necessary, explaining how they have been resolved in the revised version.
>
> - *Reliance on backtranslation may systematically disadvantage low-resource languages.*
>
> The main goal of the backtranslation quality checking pipeline is to ensure that all samples that are kept are good enough, even if this means that some low-resource languages will end up with less problems than other high-resource languages. The variation in the number of samples is a deliberate trade-off: we prioritize fewer, higher-quality samples over larger but noisier datasets, thereby maintaining consistency and reliability across languages. This approach ensures a fair multilingual comparison by preventing low-resource languages from being disadvantaged due to mistranslated samples.
>
> Furthermore, we have added Appendix B.4 with a small human audit in two languages to validate the automatic quality metrics and, by extension, the pipeline. The audit shows a strong Pearson correlation between human scoring and the automatic metric (0.8722 for Spanish and 0.8475 for French), despite differing score distributions. Crucially, the metric proves highly effective at minimizing false positives, cases where poor translations might otherwise be included in the dataset. These findings confirm consistency with human judgment and strengthen confidence in the robustness of our pipeline across languages.
>
> We are not entirely certain we have understood the reviewer’s concern correctly. If our response does not fully address the issue, we kindly invite the reviewer to clarify, and we will make every effort to respond appropriately.
>
> - *Cross-language fairness relies on filtered subsets, limited statistical testing of comparability.*
>
> In Appendix C.3, we evaluate the following null hypothesis: for a given language, the subset of correctly translated samples is as difficult as the full set. To test this, we compare accuracies of all models between the subsets of each language in English and the entire dataset (further details of this experiment can be found in the Appendix). We then restrict comparisons to languages whose subsets are statistically equivalent in difficulty to the full set (in our case, we exclude 4 of the 15 languages originally intended for benchmarking). This procedure ensures that the multilingual comparisons presented in Section 4 are fair and unbiased.
>
> - *Some models (Gemma) perform below chance; analysis of why (prompting, vision adapters) is shallow.*
>
> The below-chance performance observed for the Gemma family of models is due to formatting issues: these models **did not adhere to the required output format** specified in the prompt. We have now clarified this point in Section 4.1.
>
> - *Provide exact prompts per language and the LLM-as-judge criteria and models used.*
>
> We have added Appendix D, which **lists all the prompts used** for each language, along with the LLM-as-judge criteria and models.
>
> - *Clarify licensing and permissions for dataset redistribution.*
>
> We have a signed document from the Catalan Mathematical Society, stating that the Catalan Mathematical Society “**freely grants the use of the tests** already published for educational, academic, or similar purposes and in activities that are non-profit and without commercial effects”, as it is also publicly disclosed in Point 8 of https://scm.iec.cat/wp-content/uploads/2025/01/bases2025.pdf. If needed, we can provide a copy of this document. Our legal team has also ensured that the dataset meets all requirements to be fully released on Hugging Face. We plan to make the HF M3Kang repository publicly available immediately after the ICLR double-blind period concludes, under a license equivalent or similar to CC BY-NC-SA 4.0. At this stage, we cannot disclose the exact license document details, as doing so could compromise the double-blind review process.
>
> - *Why not include Chinese?*
>
> We have **added Chinese to the dataset**, along with many other languages, including several in non-Latin scripts (see Appendix B.1 for the full list). We initially assumed non-Latin scripts would disrupt image generation from translated content and evaluation, which rely on the Latin characters A–E for multiple-choice options. However, we recently realized that most major languages, including Chinese, follow this same format, so we have begun translating into them. We apologize for the earlier oversight. We are also working on adding Chinese to the experiments section.
>
> We trust these answers are satisfactory and resolve the ethical concerns. If so, we respectfully encourage the reviewer to consider updating the score.

---

> > ### Comment · Reviewer_HBk9 · 2025-11-26
> >
> > Thank you for your reply. I will reconsider the score after all the modifications are completed.

---

> > > ### Comment · Area_Chair_Z4P6 · 2025-11-27
> > >
> > > Dear Reviewer HBk9,
> > >
> > > We noticed that you flagged this paper for **Ethics Review** in your initial assessment. We also see that there is no explicit license statement in the submission, while the authors have provided clarification on **licensing and permissions for dataset redistribution** in their response.
> > >
> > > Could you please take another look and let us know whether the authors’ explanation addresses your concerns regarding licensing and redistribution (and, more broadly, the issues that motivated the ethics flag)? Your feedback will help us determine what further action, if any, is needed on the ethics side.
> > >
> > > Thank you for your time and assistance.
> > >
> > > Best regards,
> > >
> > > AC

---

> > > > ### Comment · Reviewer_HBk9 · 2025-11-27
> > > >
> > > > Dear AC,
> > > >
> > > > Thank you for the opportunity to clarify my concerns.
> > > >
> > > > I maintain that there are unresolved licensing issues that warrant continued ethics oversight.
> > > >
> > > > **Primary Concern - Copyright Holder:**
> > > >
> > > > The authors state they have permission from the Catalan Mathematical Society (CMS). However, to my knowledge, the copyright for Kangourou mathematics competition problems belongs to Association Kangourou sans Frontières (AKSF), the international organization that coordinates these competitions globally. The CMS is a national affiliate that administers the competition in Catalonia/Spain, but this does not necessarily grant them the authority to license the original problem content for dataset redistribution.
> > > >
> > > > I recommend the authors provide documentation of permission from the actual copyright holder.
> > > >
> > > > Best regards,
> > > >
> > > > Reviewer HBk9

---

> > > > > ### Comment · Area_Chair_Z4P6 · 2025-11-27
> > > > >
> > > > > Dear Reviewer HBk9,
> > > > >
> > > > > Thank you for the prompt and clear clarification. We acknowledge your concern that permission from the Catalan Mathematical Society may not suffice if the copyright is held by Association Kangourou sans Frontières or another central body, and we will treat the licensing issue as unresolved for now.
> > > > >
> > > > > We will follow up with the authors to request appropriate documentation or clarification from the relevant copyright holder and factor this, along with input from the ethics chairs if needed, into the final decision.
> > > > >
> > > > > Best regards,
> > > > >
> > > > > AC

---

> > > > > > ### Author Response · Authors · 2025-11-27
> > > > > > **Response by the Authors**
> > > > > >
> > > > > > We have confirmed with the Catalan Mathematical Society (SCM) that they hold ownership of the Catalan problems as well as explicit authorization from the Association Kangourou sans Frontières to grant permission for distributing the tests in Catalan for non-commercial use. This is also documented in their internal regulations, in particular, in clause 8 of document [1]. We also confirm that KSF has approved clause 8 of this document. Aligned with this clause, please note that the SCM is already publishing the Catalan problems through their website [2].
> > > > > >
> > > > > > Additionally, our legal team has conducted a very detailed analysis of the dataset’s legal terms and has approved its publication on HuggingFace. The terms are fully aligned with the licensing rights of the original data.
> > > > > >
> > > > > > [1] https://scm.iec.cat/wp-content/uploads/2025/01/bases2025.pdf. For convenience, we also provide an English translation of point 8 below:
> > > > > >
> > > > > > *8. Intellectual Property: The statements of the Cangur test are the property of the society Le Kangourou sans Frontières and, subsidiarily, the Catalan version is the property of SCM, a subsidiary of the IEC. **SCM freely grants the use of the tests already published for educational, academic, or similar purposes, and in activities that are non-profit and non-commercial. In any public use, the source must be cited**.*
> > > > > >
> > > > > > [2] https://scm.iec.cat/concurs-presencial/cangur/

---

> > > > > > > ### Author Response · Authors · 2025-11-27
> > > > > > > **Comment by the Authors**
> > > > > > >
> > > > > > > For the reviewer’s reference, all modifications to the paper have been completed. Specifically, the latest changes include:
> > > > > > > - We have extended the benchmarks to include Chinese.
> > > > > > > - We have extended the human audit to validate the quality metrics with a mid-resource language (Korean) and a low-resource language (Basque), both showing strong correlation. Results are shown in Appendix B.4.

---

> > > > > > > > ### Comment · Reviewer_HBk9 · 2025-11-28
> > > > > > > >
> > > > > > > > Thanks for the authors' response. The score is updated.

---

> > > > > > > ### Comment · Reviewer_HBk9 · 2025-11-28
> > > > > > >
> > > > > > > Thank you for the explanation. I have no concerns regarding this.

---

> ### Author Response · Authors · 2025-11-28
> **Comment by the Authors**
>
> We are deeply grateful for the reviewer’s commitment, their attention to detail in ensuring everything was correct, the thoughtful feedback provided, and the reconsideration of the score. Based on the last message, we understand that the reviewer's intention was to raise the score (though we do not know to what value). To the best of our knowledge, the entire discussion, including the reviewer’s intended score adjustment, occurred before the leak. We respectfully ask the AC to take this into consideration.

---

### Official Review · Reviewer_9m9j · 2025-10-31

**Soundness:** 3
**Presentation:** 3
**Contribution:** 3
**Rating:** 6
**Confidence:** 2

**Summary:**

This paper introduces M3Kang, a multilingual and multimodal mathematical reasoning benchmark. The dataset includes both text-only and diagram-based questions organized by grade-level difficulty. The authors developed an automated translation pipeline with backtranslation-based quality control.
Extensive benchmarking of open- and closed-source VLMs reveals key findings: models struggle with basic math and diagram-based reasoning, performance correlates with language Internet presence, Gemini-2.5-Pro leads among closed models. Additionally, direct comparison with student participants shows no significant correlation between VLM and human reasoning patterns, highlighting fundamental differences in problem-solving approaches.

**Strengths:**

1. M3Kang units multilingual, multimodal, and mathematical reasoning, enabling rigorous evaluation of VLMs.
2. The study benchmarks a diverse set of models, compares text-only vs. diagram-based performance, tests multilingual techniques, and includes human baselines, offering insights into VLM capabilities and limitations.
3. By leveraging real-world competition data and student performance, the benchmark has direct implications for educational AI development and multilingual model optimization.

**Weaknesses:**

1. The automated translation pipeline may introduce uneven quality across languages, particularly low-resource ones, and human translation (though resource-intensive) is not explored as a refinement.
2.  Without detailed classification of problem types (e.g., geometry, arithmetic, logical reasoning), it is difficult to pinpoint specific reasoning components where VLMs fail most frequently.

**Questions:**

1. Given the translation quality disparities across resource levels, have you analyzed specific error types in low-resource languages, and how might these errors confound model performance evaluations?
2. The study finds no significant correlation between VLM and human reasoning—do you hypothesize that this stems from differences in visual processing, mathematical intuition, or other factors, and how might future benchmarks better align with human problem-solving contexts?

---

> ### Author Response · Authors · 2025-11-21
> **Rebuttal by Authors**
>
> We thank the reviewer for the time spent on our submission and for the constructive remarks. Below we reproduce every point that was raised and explain how we have addressed it in the revised version.
>
> - *The automated translation pipeline may introduce uneven quality across languages, particularly low-resource ones, and human translation (though resource-intensive) is not explored as a refinement.*
>
> Automated translation is essential to reach the current scale of 108 languages. To ensure quality, we use a backtranslation-based pipeline that filters out inadequate translations, which is thoroughly explained in Section 3.3.1 and Appendix B.3. Consequently, since only translations meeting our quality standards are retained, sample counts vary across languages. This variation is a deliberate trade-off: we prioritize fewer, higher-quality samples over larger but noisier datasets, thereby maintaining consistency and reliability across languages.
>
> Furthermore, we have added Appendix B.4 with a small human audit in two languages to validate the automatic quality metrics and, by extension, the pipeline. The audit shows a strong Pearson correlation between human scoring and the automatic metric (0.8722 for Spanish and 0.8475 for French), despite differing score distributions. Crucially, the metric proves highly effective at minimizing false positives, cases where poor translations might otherwise be included in the dataset. These findings confirm consistency with human judgment and strengthen confidence in the robustness of our pipeline across languages.
>
> - *Without detailed classification of problem types (e.g., geometry, arithmetic, logical reasoning), it is difficult to pinpoint specific reasoning components where VLMs fail most frequently.*
>
> We agree that detailed classification of the problems into categories is beneficial to the interpretation of our results, possibly shedding some light on which reasoning components might give VLMs more trouble. We have performed a classification of the problems into the following five categories: geometry, algebra, arithmetic, combinatorics-probability and logic. We observe that all models consistently perform better at arithmetic and algebra, which aligns with VLMs struggling to reason with figures, as these problem types involve fewer of them. The full details of this experiment can be found in Appendix C.6.
>
> - *Given the translation quality disparities across resource levels, have you analyzed specific error types in low-resource languages, and how might these errors confound model performance evaluations?*
>
> While we did not conduct a full taxonomy of error types for low-resource languages, our back-translation pipeline was specifically designed to filter out inadequate translations, reducing the likelihood of systematic errors. This means that only translations meeting a quality threshold are retained, mitigating their impact on model evaluation. Consequently, since poor translations are excluded, they are unlikely to affect performance evaluations significantly.
>
> - *The study finds no significant correlation between VLM and human reasoning—do you hypothesize that this stems from differences in visual processing, mathematical intuition, or other factors, and how might future benchmarks better align with human problem-solving contexts?*
>
> **On the hypothesis regarding differences in visual processing, mathematical intuition, or other factors**: We are not entirely sure how to interpret this question. As noted in the paper, we already hypothesize that VLMs perform worse on vision-dependent problems because their reasoning capabilities are primarily trained on text. It is certainly true that VLMs and humans rely on fundamentally different computational architectures and learning paradigms, which inevitably shape their reasoning patterns. However, we do not believe a detailed discussion of these differences belongs in the paper. If the reviewer could clarify which specific aspects they are interested in, we would be happy to elaborate further.
>
> **On how might future benchmarks better align with human problem-solving contexts**: If we want to evaluate VLMs on more human-like problem-solving skills, future benchmarks should include multi-step reasoning tasks, combine visual interpretation with symbolic inference and assess intermediate reasoning steps rather than only final outputs. Such benchmarks would better capture the integrated reasoning strategies humans employ and provide a more meaningful measure of progress toward human-aligned capabilities. We added a comment on this in Section 4.5 of the paper, as we believe it is relevant.
>
> We trust these answers are satisfactory and respectfully encourage the reviewer to consider updating the score.

---

> > ### Comment · Reviewer_9m9j · 2025-11-26
> >
> > Thanks for the authors' response. I tend to maintain this positive score.

---

### Official Review · Reviewer_zmss · 2025-11-01

**Soundness:** 3
**Presentation:** 3
**Contribution:** 2
**Rating:** 4
**Confidence:** 4

**Summary:**

This paper presents M3Kang, a multilingual and multimodal mathematical reasoning benchmark derived from the Kangaroo Math Competition. The dataset contains 1,789 unique multiple-choice problems across 15 languages, spanning both text-only and figure-based questions. It provides a unified evaluation framework for Vision-Language Models (VLMs) in multilingual mathematical reasoning, with comparisons to over human participants. The authors design an automated three-stage pipeline: (1) extracting and cleaning Catalan problems, (2) translating them to English (M2Kang), and (3) extending to 15 languages with backtranslation-based quality filtering. Experiments benchmark major open and closed models.

**Strengths:**

(1) M3Kang fills an important gap at the intersection of multilingual, multimodal, and mathematical reasoning. Prior datasets have addressed these dimensions separately; this benchmark enables joint evaluation.

(2) The authors test about 10 VLMs, analyze correlations between accuracy and language Internet presence, compare text-only vs. figure-based questions, and benchmark multilingual reasoning methods. The analysis is thorough and supported by clear figures.

(3) Using performance data from 68,000 students allows a rare and informative human–AI comparison.

**Weaknesses:**

(1) Section 2 omits key multilingual multimodal datasets such as EXAMS-V (ACL 2024), M4U (2024), and M3Exam (NeurIPS 2024), mentioning only M5 (Schneider & Sitaram 2024) without detailed comparison. A table contrasting coverage, modality, and translation strategy would strengthen the contribution claim.

(2) The dataset originates from Catalan, chosen for data availability rather than linguistic suitability. The paper does not analyze potential bias or LLM performance limits in Catalan processing.

(3) Only the Catalan→English stage includes manual correction; multilingual translations rely solely on automatic quality metrics. A small human audit (even 100 samples) would help substantiate reliability.

**Questions:**

In the experiments comparing text-only and figure-based problems, I did not receive the full methodological details. Such comparisons should ideally be conducted on the same set of questions, where the figures in the original problems are converted into equivalent textual descriptions for the text-only version. Please clarify the full details.

---

> ### Author Response · Authors · 2025-11-21
> **Rebuttal by Authors**
>
> We thank the reviewer for the time spent on our submission and for the constructive remarks. We greatly appreciate the reviewer’s feedback and have addressed every point raised. Below, we restate each comment and explain the corresponding improvements in the revised version.
>
> - *Section 2 omits key multilingual multimodal datasets such as EXAMS-V (ACL 2024), M4U (2024), and M3Exam (NeurIPS 2024), mentioning only M5 (Schneider & Sitaram 2024) without detailed comparison. A table contrasting coverage, modality, and translation strategy would strengthen the contribution claim.*
>
> We have added the referenced sources and some others, as well as a table comparing similar datasets to ours (see **Table 1 in Section 2**). The table shows that our dataset includes by far the most languages (we cover 108 languages, more than two times the second best) and is the only one dedicated exclusively to mathematical multimodal reasoning, underscoring its value as a contribution.
>
> - *The dataset originates from Catalan, chosen for data availability rather than linguistic suitability. The paper does not analyze potential bias or LLM performance limits in Catalan processing.*
>
> To make sure there is no issue in that regard, we have now carefully **manually checked 100%** of the English samples to eliminate any potential bias from Catalan preprocessing. This led to the removal of a few instances, which are no longer part of the dataset or analysis. The resulting English dataset (M2Kang) is now linguistically accurate and fully unbiased. We have added a comment about this in Section 3.2 and a full description of the manual intervention in Appendix B.2.
>
> - *Only the Catalan→English stage includes manual correction; multilingual translations rely solely on automatic quality metrics. A small human audit (even 100 samples) would help substantiate reliability.*
>
> We have added Appendix B.4 with a **small human audit** in two languages to validate the automatic quality metrics and, by extension, the pipeline. The audit shows a strong Pearson correlation between human scoring and the automatic metric (0.8722 for Spanish and 0.8475 for French), despite differing score distributions. Crucially, the metric proves highly effective at minimizing false positives, cases where poor translations might otherwise be included in the dataset. These findings confirm consistency with human judgment and strengthen confidence in the robustness of our pipeline across languages.
>
> - *In the experiments comparing text-only and figure-based problems, I did not receive the full methodological details. Such comparisons should ideally be conducted on the same set of questions, where the figures in the original problems are converted into equivalent textual descriptions for the text-only version. Please clarify the full details.*
>
> We agree that comparing text-only and figure-based problems on the same set would provide a controlled evaluation. However, converting geometric figures into purely textual descriptions is infeasible because spatial relationships and visual cues cannot be fully captured in text without fundamentally altering the problem. To address the intent of this suggestion, we extended our Figure vs. Text-only experimental analysis by comparing VLM accuracy within difficulty levels and, separately, comparing with human performance data (Figure 7 in the paper). In the former experiment we see that VLM performs consistently better in text-only problems against problems that contain figures within every level. In the latter experiment, we see that problems that contain figures are clearly easier for humans, in contrast to VLMs, which find easier problems that are text-only. In conclusion, the results obtained with this approach strengthen our claim that multimodal questions are significantly harder for VLMs than text-only counterparts of similar difficulty, and that the difference is not attributable to easier problems. See Section 4.4 for further details.
>
> We trust these answers are satisfactory and respectfully encourage the reviewer to consider updating the score.

---

> > ### Author Response · Authors · 2025-11-27
> > **Comment by the Authors**
> >
> > We would like to inform the reviewer that we have extended the human audit to validate quality metrics using a mid-resource language (Korean) and a low-resource language (Basque), both demonstrating strong correlation. We emphasize this point as it directly addresses one of the reviewer’s main concerns. Detailed results are provided in Appendix B.4.

---

### Author Response · Authors · 2025-11-21
**Major updates and improvements**

We thank the reviewers for the useful feedback. All reviewers consistently recognized the novelty and importance of introducing a multilingual, multimodal mathematical reasoning benchmark on a common pool of problems and praised the thorough evaluation across multiple languages and VLMs.

In the corresponding sections, we have thoroughly addressed each of the reviewers’ concerns, but we would like to highlight several major changes that respond to key points raised and may be of interest to all readers here.

- We expanded the language coverage from 15 to 108 languages, now including non-Latin script languages such as Chinese, Japanese, and Arabic. The dataset now includes 111,198 problems.

- We manually checked 100% of the English samples to eliminate any potential bias from Catalan preprocessing. An explanation of this can be found in Appendix B.2.

- We conducted a small human audit in two languages to validate the automatic quality metrics and, by extension, the pipeline, showing a strong correlation between the automatic metric and human judgment. Results are shown in Appendix B.4.

- We extended our Figure vs Text-only experimental analysis, comparing the accuracy of the VLMs within levels and leveraging human data to strengthen the claim that multimodal questions are much harder for LVMs than their text-only similar-difficulty counterparts. Details are provided in Section 4.4.

- We classified the problems into categories (e.g., Algebra, Combinatorics) and performed an analysis based on this classification, detailed in Appendix C.6.

---

> ### Author Response · Authors · 2025-11-27
> **Additional updates and improvements**
>
> We made further improvements during the final revision:
> - We have extended the human audit to validate the quality metrics with a mid-resource language (Korean) and a low-resource language (Basque), both showing strong correlation. Results are shown in Appendix B.4.
> - We have extended the benchmarks to include Chinese.

---

### Author Response · Authors · 2025-11-28
**Comment for the AC**

Dear AC,

We sincerely appreciate all the reviewers' dedication, attention to detail, and thoughtful feedback. We believe they provided excellent comments that helped further improve the quality of our paper. We understand that, due to the OpenReview leak incident, the review process has been paused. We would like to respectfully highlight that reviewer HBk9 had agreed to update the score. Additionally, we were hoping other reviewers might consider revising their evaluations, as we have thoroughly addressed all comments and significantly improved both the paper and the M3Kang dataset based on their feedback.

---

### Meta-Review · Area_Chair_M3qp · 2026-01-02

**Summary:**

This paper presents M3Kang, a multimodal reasoning dataset for evaluating the multilingual mathematical reasoning abilities of state-of-the-art vision–language models, built on problems from the Kangaroo Math Competition. The paper extends the findings of Cherian et al. (2024) by exploring how multilinguality affects reasoning. To this end, it translates question text from the original Catalan language into English and then into 108 different languages using a proposed mixed automatic/manual pipeline. Experiments on state-of-the-art closed- and open-source VLMs show marked performance differences between high-resource and low-resource languages, with most models performing best in English, and also demonstrate that models struggle with multimodal reasoning.

**AC Comments:**

The paper received mixed reviews initially, including one borderline reject, one reject, and two borderline accepts, with one reviewer indicating a willingness to raise the score from reject to accept. Reviewers raised several concerns, namely: i) reliance on backtranslation as the primary approach for translating problems from English into 108 other languages sourced from the original Catalan; ii) lack of positioning of the proposed dataset within the context of other multilingual datasets; iii) lack of insights into the types of errors produced during VLM reasoning; iv) the confounding nature of the results when multilinguality and multimodality are intermixed; and v) issues regarding the rights to release the dataset.

The AC finds that the authors provided a strong rebuttal addressing many of these concerns, including a partial quality check of the backtranslation on a small subset, the addition of Table 1 positioning the dataset among existing datasets, analysis of VLM classification errors in the English language, and clarification of the agreement to release the dataset. After considering the entirety of the arguments, the AC believes the paper introduces a useful dataset for multilingual mathematical problem solving. However, several issues remain that blur the key takeaways.

Specifically:

i) The key issue addressed in the paper is multilingual problem solving, yet it is unclear why multilinguality and multimodality must be considered jointly for mathematical reasoning. It remains ambiguous whether multilingual and multimodal reasoning need to be analyzed together, or whether they are separable challenges. The main conclusion—that VLMs struggle with problems involving both text and images—is already known from prior work (e.g., MathVerse, Zhang et al., 2025, SMART-840, Cherian et al., 2024). However, the effects of multilinguality, which are the focus of this paper, are not sufficiently isolated to clearly understand what models struggle with when problems are translated. It is unclear whether the observed degradation is due to translation errors, limited exposure to low-resource languages, or deficiencies in mathematical reasoning itself versus language understanding. As a result, the takeaways regarding multilinguality are not articulated with sufficient clarity.

ii) The paper positions the proposed dataset in Table 1 as one of the largest among related datasets, but does not clarify how many problems in other datasets are in their original languages versus automatically translated. In principle, any English dataset could be automatically translated into multiple languages, but doing so requires careful quality assurance beyond automated metrics or small-scale human studies. Greater emphasis is therefore needed on validating translation quality. In this context, the AC questions whether it would be preferable to separate multimodality and multilinguality, for example by focusing solely on text-based problems or by translating an existing dataset (e.g., GSM8K) into multiple languages to study multilingual mathematical reasoning in isolation.

iii) The results in Figure 18, which analyze classification performance across problem types, consider only the English language and do not analyze how multilinguality affects these categories. An important advantage of the Kangaroo competition is the availability of data from human participants; however, this data appears to come only from students who participated in the competition in the Catalan language (if the AC is not mistaken). Since the Kangaroo competition is held in many other languages, it may be possible to obtain original competition materials in those languages, enabling multilingual analysis without relying entirely on backtranslation unless translations are exhaustively verified for accuracy.

In summary, while the questions posed in this paper are interesting, the current version lacks sufficient insight into the role of multilinguality in multimodal mathematical reasoning. As such, the AC believes the paper is not yet ready for acceptance.

**Reviewer Concerns:**

*Reviewer zmss* points out that the paper omits prior multilingual math reasoning datasets (EXAMS-V, ACL 2024; M4U, 2024; and M3Exam, NeurIPS 2024) from the related work. The reviewer also seeks clarity on why Catalan was chosen as the source language, the ability of LLMs to process this language, and notes that translations into the various target languages rely on automatic quality metrics, potentially overlooking translation errors in the performance analysis.

*Reviewer 9m9j* raises similar concerns, including the presence of translation errors that could influence performance and the lack of human validation. The reviewer also emphasizes the need to analyze errors across problem types to better understand the impact of the multilingual setup, and questions the paper’s conclusions regarding the lack of correlation between human and VLM reasoning.

*Reviewer HBk9* echoes these concerns, highlighting the over-reliance on backtranslation for generating problems in other languages, which may favor high-resource languages, the lack of reliable assurance of translation quality across the full dataset, and the absence of Chinese languages in the study. The reviewer also seeks clarification regarding the rights to use proprietary problems from the Olympiad.

*Reviewer yh29* questions the reported differences in performance between multimodal and text-only problems.

**Reviewer Scores:**

**Reviewer zmss:** The authors revised the paper by adding Table 1, which compares the proposed dataset with other related multilingual datasets and highlights that it covers more languages (108) than prior datasets (e.g., 41 in M5), with problems specific to mathematical reasoning. The authors also confirm that translations from Catalan to English were manually checked for errors, and that a small-scale human audit was conducted to assess the correctness of translations into other languages.

[*AC’s take on the response*] The new Table 1 is useful for positioning the dataset among other multilingual datasets. However, the AC believes the table may also be somewhat misleading, as the claimed 111K problems were generated by automatically translating 1,747 original problems, while it is not specified how many original problems were used in the other datasets. Although a subset of translated problems was checked for errors, a large portion of the dataset remains unverified. As such, the AC believes the authors’ response only partially addresses the reviewer’s concerns.

**Reviewer 9m9j:** As suggested by the reviewer, the authors added Figure 18, which breaks down VLM performance across five problem categories using the English translations. However, this analysis does not account for the multilingual aspect of problem solving. Regarding deeper insights into which aspects of multilinguality the models struggle with, the authors acknowledge that such analysis was not conducted. Concerning the broader question of correlations between human and VLM reasoning, the authors consider this beyond the scope of the study, while acknowledging that visual reasoning is a key bottleneck. The reviewer tends to maintain the score at “borderline accept.”

**Reviewer HBk9:** The authors argue that backtranslation is used to ensure quality, and report a human audit on a small subset of translations showing a correlation score greater than 0.8 between human and automatic evaluations. The authors also confirm that they have obtained the necessary rights to use and publicly release the dataset, and note that Chinese has been added to the list of considered languages.

[*AC’s take on the response*] The reviewer appears to be satisfied with the response and seems to have raised the score.

**Reviewer yh29:** The authors confirm that VLMs perform better on text-only problems than on problems that include figures.

---

### Decision · Program_Chairs · 2026-01-26

Reject